



# The contribution of shipping to air pollution in the Mediterranean region – a multimodel evaluation: Comparison of photooxidants NO₂ and O₃

Lea Fink[1], Matthias Karl[1], Volker Matthias[1], Sonia Oppo[2], Richard Kranenburg[3], Jeroen Kuenen[3], Jana Moldanova[4], Sara Jutterström[4], Jukka-Pekka Jalkanen[5], Elisa Majamäki[5]

[1]Helmholtz-Centre Hereon, Institute of Coastal Environmental Chemistry, 21502 Geesthacht, Germany
[2]AtmoSud, Air Quality Observatory in the Provence-Alpes-Côte d'Azur region, 13006 Marseille, France
[3]TNO, Netherlands Organization for Applied Scientific Research, 3584 CB Utrecht, The Netherlands
[4]IVL, Swedish Environmental Research Institute, 411 33 Göteborg, Sweden
[5]FMI, Finnish Meteorological Institute, FI-00560 Helsinki, Finland

*Correspondence to*: Lea Fink (lea.fink@hereon.de)

**Abstract.** Shipping has a significant contribution to the emissions of air pollutants such as $NO_x$ and particulate matter (PM), and the global maritime transport volumes are projected to increase further in the future. The Mediterranean Sea contains the major route for short sea shipping within Europe and contains the main shipping route between Europe and East Asia. Thus, it is a highly frequented shipping area, and high levels of air pollutants with significant contributions from shipping emissions are observed at monitoring stations in many cities along the Mediterranean coast.

The present study is part of the EU H2020 project SCIPPER (Shipping contribution to Inland Pollution Push for the Enforcement of Regulations). Five different regional chemistry transport models (CAMx, CHIMERE, CMAQ, EMEP, LOTOS-EUROS) were used to simulate the transport, chemical transformation and fate of atmospheric pollutants in the Mediterranean Sea for 2015. Shipping emissions were calculated with STEAM version 3.3.0, and land-based emissions were taken from the CAMS-REG v2.2.1 dataset for a domain covering the Mediterranean Sea on a resolution of 12x12 km² (or 0.1° x 0.1°). All models used their standard setup for further input. Ship contribution was calculated with the zero-out method. One run using the tagging method was performed with LOTOS-EUROS. The model outputs were compared against each other and to measured background data at monitoring stations.

The results showed differing outputs regarding the time series and pattern of model outputs but similar results with regard to the overall underestimation of $NO_2$ and overestimation of $O_3$. The contribution from ships to the total $NO_2$ concentration was especially high at the main shipping routes and coastal regions (25% to 85%). The contribution from ships to the total $O_3$ concentration was lowest in regions with the highest $NO_2$ contribution (down to -20%). A comparison of the zero-out and tagging methods has shown that the annual mean ship contribution to the total $NO_2$ concentration is smaller (up to 75%) and has a lower range when the tagging method is used. CAMx and CHIMERE simulated the highest ship contributions to the $NO_2$ and $O_3$ air concentrations. Additionally, the strongest correlation was found between CAMx and CHIMERE, which can be traced back to the usage of the same meteorological input data. The CMAQ, EMEP and LOTOS-EUROS simulated values





were within one range for the $NO_2$ and $O_3$ air concentrations. Regarding deposition output, larger differences between the models were found when compared to air concentration. These uncertainties and deviations between models are caused by

deposition mechanisms, which are unique within each model. A reliable output from models simulating ship contributions can be expected for air concentrations of $NO_2$ and $O_3$.

## 1 Introduction

Shipping activity and freight transport via ships are growing, and previous studies have shown that the relative contribution from shipping to total air pollution will also increase (Brandt et al., 2013). Once in the atmosphere, these emissions are

transported over several hundreds of kilometers, with 70 % of shipping emissions occurring less than 400 km from the coast (Eyring et al., 2010; Endresen et al., 2003). Several previous studies have pointed out the negative effect of shipping emissions on the concentration of air pollutants, playing a role as greenhouse gases, impacting human health or contributing to acidification and eutrophication (Tysro and Berge, 1997; Corbett and Fischbeck, 1997; Corbett et al., 1999).

Nevertheless, maritime transport plays a vital role in the international trade of goods worldwide as well as in the European

Union (EU). The Eurostat Press Office (2016) stated that for 2015, the value of EU trade of goods with non-EU countries transported by the sea was approximately 51% of EU traded goods. The Mediterranean Sea contains one main shipping route between Europe and Asia, being the region in Europe with maximal contribution from shipping emissions to gaseous pollutants, in addition to the North Sea (Viana et al., 2014).

Additionally, as one of the fastest growing sources of greenhouse gas emissions, shipping emissions directly result in health

problems and have adverse effects on ecosystems (Brandt et al., 2013). The wide range of gaseous pollutants, such as nitrogen oxides ($NO_x = NO_2 + NO$), coming from shipping emissions have negative impacts by forming smog and acid rain and contribute to eutrophication (Jägerbrand et al., 2019; Brandt et al., 2013; Karl et al., 2019b; Matthias et al., 2010).

Moreover, $NO_x$, as a primary pollutant, plays an important role in the formation of $O_3$ and in the deposition of reactive nitrogen compounds (Eyring et al., 2010). The oxidation of VOCs (volatile organic compounds) produces ozone in the troposphere

when $NO_x$ and sunlight are present. $O_3$ can inflame and damage the respiratory system, make the lungs more susceptible to infection and intensify lung diseases (EPA, 2021). Although it is not directly emitted, $O_3$ is an important compound in photochemistry. Especially in the Mediterranean Sea during summer, when radiation is high, the contribution of shipping emissions to mean surface $O_3$ concentrations can be significant (Aksoyoglu et al., 2016).

Atmospheric nitrogen deposition mainly comes from agricultural activities and combustion processes such as those in shipping

(Aksoyoglu et al., 2016). This increase in bioavailable nitrogen deposition causes eutrophication (Jägerbrand et al., 2019). The deposition of $O_3$ affects the plant's stomata, damages the plants, changes water and carbon cycling and reduces crop yields (Clifton et al., 2020).





Chemistry transport models (CTMs) can be applied to simulate the transport of air pollutants as well as chemical transformation
and deposition. These models can be used at different scales, depending on the domain they cover and the question to be
answered.

Although shipping emissions have a significant impact on air pollution by $NO_2$ in the Mediterranean Sea (Marmer and
Langmann, 2005), few regional-scale chemistry transport modeling studies have focused on this domain. A literature review
study focusing on the assessment of the impacts of shipping emissions on air quality in European coastal areas by Viana et al.
(2014) showed that studies regarding shipping emissions in the Mediterranean Sea emphasize $PM_x$ levels and their chemical
composition instead of gaseous pollutants. Marmer and Langmann (2005) investigated the Mediterranean Sea, but on a larger
scale or without the comparison of different CTMs. Other studies focus on smaller domains over the Iberian Peninsula
(Baldasano et al., 2011; Nunes et al., 2020), the eastern part of the Mediterranean Sea with the Arabian Peninsula (Večeřa et
al., 2008; Tadic et al., 2020; Celik et al; Friedrich et al., 2021) or urban scale and harbor cities (Schembari et al., 2012; Donateo
et al., 2014; Prati et al., 2015). However, none of these studies modeled the ship contribution on a regional scale with a
subsequent model comparison of different chemical transport models. A comparison of outputs of regional-scale chemistry
transport models was performed for the Baltic Sea or for all of Europe (Karl et al., 2019a; Im et al., 2015a) but not exclusively
for the western Mediterranean region.

Dry deposition is a substantial sink for atmospheric pollutants. Furthermore, it determines the net flux of pollutants to the
Earth's surface (Galmarini et al., 2021). Accurate estimates of dry deposition are required for reliable predictions of
atmospheric concentrations, since it is an important loss process scaling with concentrations close to the ground (Emerson et
al., 2020; Vivanco et al., 2018). $NO_2$ deposition contributes to eutrophication, followed by biodiversity loss, whereas $O_3$ dry
deposition injures plant tissues and reduces plant productivity (Vivanco et al., 2018; Clifton et al., 2020). The deposition of N
and S was investigated in previous studies (i.e., Vivanco et al., 2018; Jutterström et al., 2021; Galmarini et al., 2021).
Nevertheless, few studies have performed model intercomparison for dry deposition; thus far, none of the studies have focused
on ship impact over the western part of the Mediterranean Sea. Comparing the dry deposition mechanisms of different models
is essential since these mechanisms are unique for each model. In Galmarini et al. (2021), deposition schemes of different
models were compared, including LOTOS-EUROS and CMAQ, which are also part of the present study. They showed, i.e.,
Differences in surface resistance calculation and deposition pathways: LOTOS-EUROS uses a single deposition pathway to
soil. In comparison, CMAQ uses two deposition pathways for deposition to soil (one for vegetation-covered and one for bare
soil).

Additionally, another important factor is the land use-land cover (LUCL) on which dry deposition strongly depends but is
unique in each model. This was also stated by Vivano et al. (2018), explaining that even if models apply similar algorithms in
their deposition schemes, they may use different land use or leaf index area data. Thus, mainly over land areas, differences in
model outputs are to be expected. A similar mechanism and model output for dry deposition is expected over water and
therefore over most of the considered domain in the present study.

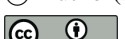



The Ship Traffic Emission Assessment Model (STEAM) has been previously applied to evaluate shipping emissions in different regions, such as the North Sea or Baltic Sea (Jalkanen et al., 2009; Jonson et al., 2015; Aulinger et al., 2016; Barregard et al., 2019) or the Iberian Peninsula (Nunes et al., 2020), as well as in European (Jalkanen et al., 2016) and global regions (Johansson et al., 2017). However, the model has not been previously used in a study focusing entirely on the western

Mediterranean Sea region.

In addition, the Mediterranean Sea is not yet the ECA (Emission Control Area). The contracting parties of the Barcelona Convention agreed to designate the Mediterranean Sea as an Emission Control Area for Sulfur emissions (MedECA) by 2025. Nevertheless, although $SO_2$ emissions must be reduced by 50 % to 80 % by 2030, $NO_x$ emissions from ships will grow without further control and likely exceed emissions from land-based sources in the European Union after 2030 (Cofala et al., 2018).

Furthermore, the current state of air pollution is calculated to have a basis for investigating the effects of additional legislation. It is important to simulate the ship contribution to several air pollutants to show the impact of ships in a larger area.

The Horizon 2020 SCIPPER project (Shipping Contributions to Inland Pollution Push for the Enforcement of Regulations) aims to determine how existing regulations ensure compliance with the legislation on emissions to air from ships. One part of this project was to focus on CTMs and their possible supportive effects in the monitoring of compliance of threshold levels.

The present study compares and evaluates five different CTMs concerning their predictions of the dispersion and transformation of air pollutants. The main focus of this study is to compare the output of models regarding the ship contribution to atmospheric concentrations and dry deposition of $NO_2$ and $O_3$. Using this comparison, important differences in the photochemical processing between the models and the balance of photochemistry in the models focusing on shipping will be highlighted. Furthermore, the model performance was quantified by comparing the modeled data against the measured data of

air pollutants at background stations in coastal areas of the Mediterranean Sea. The performance of the models was compared based on statistical indicators.

By using five different CTMs in this part of the SCIPPER project, a more robust estimate of the ship contribution to the air pollution can be given. To date, the present study is the first multimodel study to compare ship contributions to five regional-scale CTMs in the Mediterranean Sea.

**2 Materials and Methods**

**2.1 Models**

Five different regional-scale CTMs were used for this study, run by four institutions: CAMx and CHIMERE by AtmoSud, CMAQ by Helmoltz Centre Hereon, EMEP by IVL Swedish Environmental Research Institute and LOTOS-EUROS by TNO Netherlands Organization for applied scientific research.

The goal was to have a model setup as similar as possible for all models to receive comparable outputs. As a base, an inner and outer domain with grid resolution was established. Additionally, the emissions were provided for one year. Especially of importance in the present study was the method for calculating the ship contribution.



An overview of the input data is shown in Table 1. Input data were the same for shipping emissions using STEAM (version 3.3.0.; Jalkanen et al., 2009; Jalkanen et al., 2012; Johansson et al., 2013; Johansson et al., 2017), land-based emissions

(CAMS-REG, v2.0) as well as projection (WGS84_lonlat), domain (Mediterranean Sea), resolution (0.1° x 0.1°, 12 x 12 km) and the modeled year (2015). Input data were different for meteorological input data, boundary and initial conditions because the CTMs used their standard setup.

The output of the model runs should all contain $NO_2$ and $O_3$ in µg/m³ at an hourly resolution on a 2D grid from the lowest layer and be provided as a netcdf file following CF conventions. The lowest layer on the ground was used in the present study.

With all models, a reference run for the current air quality situation was performed, including all emissions (base case). Furthermore, all models did one run without the emissions from shipping (noship case). The difference between the calculations with all emissions and the calculation without shipping emissions is used to determine the contribution of ships to the ambient pollutant concentration (zero-out method). This was done for all five models.

One run was performed with the tagging method by LOTOS-EUROS. In the tagging method, emitted species are tagged

according to their emission source. These are not necessarily sectors but can also be countries, regions, time of emission, etc. These tags are transferred to other species during the subsequent chemical reactions, where conserved atoms C, N and S are tracked throughout the chemical calculations. Because $O_3$ is not directly emitted, the tagging method cannot be used directly to tag $O_3$, so the tagging method is not applied for ozone in this study. This results in a model calculation with identical chemical behavior, while zero-out methods change the chemical behavior of the model. In this study, a tag was placed on the

shipping emissions to obtain the shipping contribution of the current chemical regime. For a comparison of the different ship contribution methods, LOTOS-EUROS also performed a run with the zero-out method.

**Table 1: Main model parameters and input data for the five chemical transport models.**

| Model parameter | CAMx | CHIMERE | CMAQ | EMEP | LOTOS-EUROS |
|---|---|---|---|---|---|
| Grid resolution inner domain | 12x12 km² | 12x12 km² | 12x12 km² | 0.1°x 0.1° | 0.1°x 0.1° |
| Grid resolution outer domain | 36x36 km² | 36x36 km² | 36x36 km² | none | 0.5°x 0.25° |
| Land-based emissions | CAMS-REG | CAMS-REG | CAMS-REG | CAMS-REG | CAMS-REG |
| Shipping emissions | STEAM | STEAM | STEAM | STEAM | STEAM |
| Meteorological driver | WPS/WRF | WPS/WRF | COSMO-5 CLM | ECMWF (IFS) | ECWMF (IFS) |



| **Boundary conditions** | Mozart-4 output | Gaseaous species: LMDz-INCA model  Aerosols: GOCART model | IFS_CAMS cycle45r1 | boundary conditions provided with the open source model distribution for year 2015 | CAMS C-IFS |
| --- | --- | --- | --- | --- | --- |

### 2.1.1 Model description CAMx

CAMx (Comprehensive Air Quality Model with Extensions) is a Eulerian photochemical dispersion model developed by
Ramboll Environ. Version CAMx v6.50 of the model was used in the present study.

For this study, a first domain with a 36 km resolution was defined at the European scale. A second nested domain was defined, named MEDI12 (147x249 points), and covered the center of Europe with a resolution of 12 km. Both meteorological and chemical transport simulations were provided for these domains. WRFv3.9 was run for the simulation of meteorological conditions with 28 vertical layers up to 50 hPa, with FNL data for initial conditions.

For the CAMx simulation, boundary conditions from the Mozart-4 output and the PSAT and OSAT modules (Particulate Source Apportionment Technology and Ozone Source Apportionment Technology) were activated to quantify the aerosol and ozone sources in Europe and especially the contribution from maritime emissions.

The gas phase chemical mechanism is CB05, in which the NMVOC emissions are split into 13 species (TERP, ISOP, XYL, TOL, ETOH, MEOH, IOLE, OLE, ETH, ALD2, PAR, ETHA and FORM) and describe approximately 156 reactions. For
semivolatile inorganic species (sulfate, nitrate, and ammonium), the equilibrium concentration is calculated using the thermodynamic model ISORROPIA (Nenes et al., 1998). Fourteen vertical levels are simulated with a first layer height of approximately 10 km.

### 2.1.2 Model description Chimere

CHIMERE is an offline chemistry transport model developed by LMD-IPSL/CNRS (Menut et al., 2013). The
CHIMERE2017r4 version of the model was used in this study.

WRFv3.9 (Weather Research and Forecasting Model) was run for the simulation of meteorological conditions with 28 vertical layers up to 5 0 hPa, with FNL data for initial conditions.

Concerning CHIMERE simulation, boundary conditions are monthly mean climatologies taken from the LMDz-INCA model (Laboratoire de Météorologie Dynamique General Circulation Model – INteraction with Chemistry and Aerosols; Schultz et
al., 2006) for gaseous species and from the GOCART model (Global zone Chemistry Aerosol Radiation and Transport; Ginoux et al., 2001) for aerosols (desert dust, carbonaceous species and sulfate). The gas phase chemical mechanism is MELCHIOR2 (Modele Lagrangien de Chimie de l'Ozone a l'echelle Regionale), in which the NMVOC emissions are split into 10 species (C2H6, NC4H10, C2H4, C3H6, C5H8, OXYL, HCHO, CH3CHO, CH3COE and APINEN) and describe approximately 120 reactions. For semivolatile inorganic species (sulfate, nitrate, and ammonium), the equilibrium concentration is calculated



using the thermodynamic model ISORROPIA (Nenes et al., 1998). Nine vertical levels are selected with a first layer height at 20 m to 25 m. Sea salt emissions were calculated as described in Monahan, 1986. MEGAN Model v2.04 calculated biogenic emissions separately (Guenther et al., 2006), which were then included in the land-based emissions

For this study, a first domain with a 36 km resolution at the European scale was defined. A second domain was nested within, named MEDI12 (147 x 249 points), and covered the center of Europe with a resolution of 12 km. Both meteorological and

chemical transport simulations were provided for these domains.

### 2.1.3 Model description CMAQ

The CMAQ Model v5.2 with the aero6 model calculates on the basis of emission input data air concentration as well as deposition fluxes of atmospheric gases and aerosols (Byun and Schere, 2006; Appel et al., 2017). Atmospheric chemistry is used by the Carbon Bond V mechanism (Yarwood et al., 2005) cb05tucl with updated toluene chemistry (Whitten et al., 2010),

including the chlorine chemistry extension (CB05-TUCL; https://www.airqualitymodeling.org/index.php/CMAQv5.0_Chemistry_Notes, accessed May 2021). The aerosol scheme AERO6 is used for the formation of secondary inorganic aerosols. Sulfuric acid ($H_2SO_4$), nitric acid ($HNO_3$), hydrochloric acid (HCl) and ammonia ($NH_3$) gas phase – aerosol partition equilibrium is solved by the ISORROPIA mechanism (Fountoukis and Nenes, 2007; Nenes et al., 1998). Contained within is the formation of secondary organic aerosol (SOA) from isoprene,

terpenes, benzene, toluene, xylene and alkanes (Carlton et al., 2010; Pye and Pouliot, 2012).

Sea salt emissions were calculated as described in Kelly et al. (2010). Biogenic emissions (NMVOC from vegetation and soil NO) were calculated separately with the MEGAN Model v3 (Model of Emissions of Gases and Aerosols from Nature; Guenther et al., 2012). Emissions of windblown dust were not considered. CMAQ Models 30 vertical layers, with the lowest layer from 0 m to 42 m and the second layer from 42 m to 85 m.

The COSMO model simulated the meteorological data for CMAQ, applying the version COSMO5-CLM16 (Schultze and Rockel, 2018; Petrik et al., 2021). The MCIP (Meteorology-Chemistry Interface Processor) processed meteorological model output into the input format required for CMAQ. The vertical resolution of the meteorological model output was 40 terrain-following geometric height levels up to 22 km. The Boundary Condition driver used was IFS-CAMS cycle45r1 (Integrated Forecasting System – Copernicus Atmosphere Monitoring Service; Inness et al., 2019) with a vertical resolution of 60 sigma

levels up to 65 km.

To prevent the effects from initial conditions on the simulated atmospheric concentrations in 2015, the model run started with a spin up run in mid-December 2014. The grid size of the Mediterranean Sea domain was 12 x 12 km², nested in a 36 x 36 km² domain covering all of Europe.

### 2.1.4 Model description EMEP

The EMEP MSC-W (European Monitoring and Evaluation Programme, Meteorological Synthesizing Centre – West, https://www.emep.int/mscw/index.html, assessed June 2021) model is a limited area, terrain-following hybrid coordinate



model designed to calculate air concentrations and deposition fields for major acidifying and eutrophying pollutants, photooxidants and particulate matter (Simpson et al., 2012; Simpson et al., 2020).

In this study, a 0.1° x 0.1° resolution grid on long–lat projection and with 20 vertical levels was used. The meteorological input data are based on forecast experiment runs with the Integrated Forecast System (IFS), a global operational forecasting model from the European Centre for Medium-Range Weather Forecasts (ECMWF). The meteorological fields are retrieved on 0.1° x 0.1° long–lat coordinates. Vertically, the fields on 60 eta (η) levels from the IFS model are interpolated onto the 20 EMEP eta levels.

The model version used was rv4.34 with chemical mechanism EmChem 19a (Simpson et al., 2012; Simpson et al., 2020). The mechanism builds on surrogate VOC species (Simpson et al., 2012; extended with benzene and toluene) and has 171 gas phase and heterogeneous reactions. The model always assumes equilibrium between the gas and aerosol phases using the MARS equilibrium module (Model for an Aerosol Reacting System) of Binkowski and Shankar (1995). For secondary organic aerosol (SOA), a so-called volatility basis set (VBS) approach (Robinson et al., 2007; Donahue et al., 2009; Bergström et al., 2012) is used. All primary organic aerosol (POA) emissions are treated as nonvolatile to keep emission totals of both PM and VOC components the same as in the official emission inventories, while the semivolatile ASOA and BSOA species are assumed to oxidize (age) in the atmosphere by OH reactions (Simpson et al., 2012).

The following natural emissions are calculated in the model for each grid cell and at every model time step: Biogenic emissions of isoprene and monoterpenes use near-surface air temperature and photosynthetically active radiation. Soil NO emissions from soils of seminatural ecosystems are specified as a function of N deposition and temperature. The generation of sea salt aerosol over oceans is driven by the surface wind, and the EMEP model's parameterization scheme for calculating sea salt generation is based on two source functions, those of Monahan et al. (1986) and Mårtensson et al. (2003). The key parameter driving dust emissions is wind friction velocity. Additionally, daily emissions from forest and vegetation fires are taken from the "Fire INventory from NCAR version 1.0" (FINNv1; Wiedinmyer et al., 2011). For this study, the initial and boundary conditions provided with the open source model distribution for 2015 were used.

### 2.1.5 Model description LOTOS-EUROS

LOTOS-EUROS is a Eulerian chemistry transport model (Manders et al., 2017). The model simulates air pollution in the lower troposphere and is of intermediate complexity, allowing ensemble-based simulations and assimilation studies. LOTOS-EUROS performs hourly calculations using ECMWF (European Centre for Medium-Range Weather Forecasts) meteorological data. The gas phase chemistry follows the TNO CBM-IV scheme (Schaap et al., 2008). The dry deposition fluxes are calculated with the Deposition of Acidifying Compounds (DEPAC) 3.11 module, following the resistance approach, which includes a calculation of bidirectional NH₃ fluxes (van Zanten et al., 2010; Wichink Kruit et al., 2012). The wet deposition fluxes are computed using the CAMx approach, which includes both in-cloud and below-cloud scavenging (Banzhaf et al., 2012). LOTOS-EUROS has a dynamical vertical layer structure with 5 layers in total. The first layer is at 25 m, while the second layer follows the meteorological boundary layer. On top of that, up to 3500 m and one top layer up to 5000 m above sea level



two evenly distributed reservoir layers are defined. The model has participated in multiple model intercomparison studies (Bessagnet et al., 2016; Colette et al., 2017), showing overall good performance.

## 2.2 Model Domains and Nesting

The domain for the intercomparison of the western part of the Mediterranean Sea covered a spatial extent from longitude: -1.0° to 31.2° and latitude: 32.8° to 46.8°. The grid cell size used was 12 x 12 km² interpolated on a 0.1° x 0.1° grid nested in

a larger 36 x 36 km² grid (except EMEP) covering all of Europe, as shown in Figure 1.

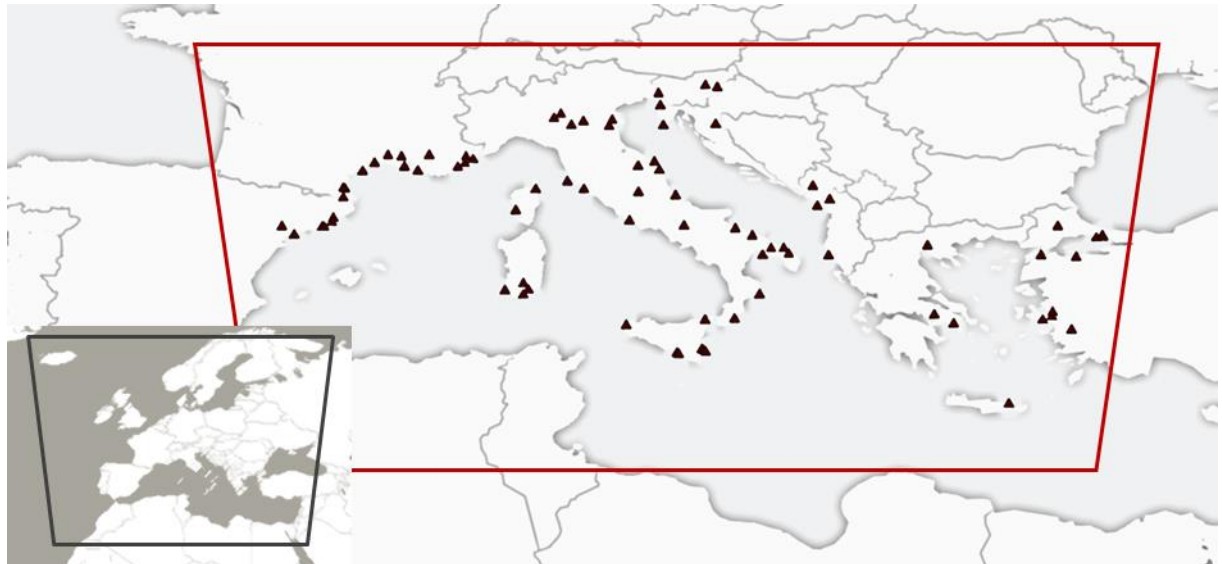

**Figure 1: Domains and measurement stations. Red trapeze displays the 12 x 12 km² domain, black triangles are locations of measurement stations. On bottom left the larger 36 x 36 km² domain is displayed.**





### 2.3 Emissions

#### 2.3.1 Land-based Emissions

Annual anthropogenic land-based gridded emissions for 2015 obtained from the CAMS-REG v2.2 emission inventory were used as input by all five compared models. Gridded emission files contain GNFR (Gridded Nomenclature for Reporting) emission sectors for each country for the air pollutants $NO_x$, $SO_2$, NMVOC, $NH_3$, CO, $PM_{10}$, $PM_{2.5}$, and $CH_4$. The emissions are provided at a spatial resolution of $1/10°$ x $1/20°$ in longitude and latitude (i.e., ~ 6 x 6 km over central Europe).

The height distribution of emissions per GNFR sector was determined as described in Bieser et al. (2011b). The temporal 255 distribution was determined by separating the annual emissions of each sector into hourly emission data with data splitting as described in Granier et al. (2019). PM was split as described in Bieser et al. (2011a), $NO_x$ was split according to Manders-Groot et al. (2016), and NMVOC split was used as provided for the CAMS-REG v2.2 emission inventory (Granier et al., 2019).

#### 2.3.1 Shipping Emissions

The shipping emission dataset produced with the STEAM model has a spatial resolution of 12 x 12 km² and a temporal resolution of 1 hour. The STEAM emissions are divided into two vertical layers (0 m to 36 m; 36 m to 1000 m) and are provided for mineral ash, carbon monoxide (CO), carbon dioxide ($CO_2$), elemental carbon (EC), $NO_x$, organic carbon (OC), $PM_{2.5}$, particle number count (PNC), sulfate ($SO_4$), $SO_x$ (containing $SO_2$ and $SO_3$) and VOC. To reduce the number of generated emission maps and the computational resources needed to run the STEAM model, VOC emissions were divided into four 265 categories according to their properties as a function of the engine load. Emission factors for VOC are based on the average values taken from various publications (Agrawal et al., 2008; Agrawal et al., 2010; Sippula et al., 2014; Reichle et al., 2015).

In CAMx, all shipping emissions are put in the first layer. For CHIMERE, all shipping emissions above 36 m and 88 % of the emissions below 36 m have been added to the second layer. Only 12 % of the emissions below 36 m were emitted in the first layer of the model. This was calculated based on the STEAM emission dataset and therein contained stack heights. 270 Additionally, in CMAQ, shipping emissions were distributed in the two lowest layers, emissions below 36 m were attributed to the lowest layer, and emissions above 36 m were in the second layer. For EMEP simulations, the STEAM emissions were summed from hourly to daily emissions and attributed to the lowest layer (up to 90 m). In LOTOS-EUROS, emissions below 36 m are assigned ~ 70 % to the first layer, which is 25 m thick, and ~ 30 % to the second layer. Emissions above 36 m are divided over different height classes 30 % between 36 m and 90 m, 30 % between 90 m and 170 m, 30 % between 170 m to 275 310 m and 10 % between 310 cm and 470 m. Due to the dynamic second model layer (following the meteorological boundary layer), those emissions are put in the second and/or third model layer. In the case of a well-mixed and vertically extended meteorological boundary layer (above 470 m), all emissions are in this second layer, whereas when the boundary layer is shallow, some emissions are put in the third layer.





## 2.4 Deposition Mechanisms

Deposition velocities for gaseous species in CHIMERE, CMAQ and LOTOS-EUROS are based on the formula introduced by Wesely (1989). This formula is the reciprocal sum of aerodynamic resistance ($R_a$), quasi-laminar sublayer resistance ($R_b$) and surface resistance ($R_c$). Nevertheless, all models differ in calculating the single variables. $R_a$ depends on meteorology and surface roughness, which is model dependent. $R_b$ is determined by the friction velocity, depending on the surface type. $R_c$ is the bulk surface resistance, containing different components, i.e., leaf stomata, soil, leaf litter, etc. All of these components

use input data that are unique for each model.

In CHIMERE, $R_b$ is estimated following Hicks et al. (1987). The resistance $R_c$ formulation follows Erisman et al. (1994) and the developments made in the EMEP model (Emberson et al., 2000; Simpson et al., 2003; Simpson et al., 2012). It uses a variety of additional resistances, mostly to account for stomatal and surface processes, both of which are depending on the land use type and season. In CMAQ, the m3dry mechanism was used, which takes $R_a$ and $R_b$ from the provided meteorological

data. $R_c$ is calculated in CMAQ as described in Pleim and Ran (2011). The resistance $R_b$ in LOTOS-EUROS is described following the EDACS system (Erisman et al., 1994). In van Zanten et al. (2010), the parametrizations of different resistances $R_c$ that contribute to resistance for dry deposition of $NO_2$ and $O_3$ are described, depending on land use type.

CAMx uses the gas resistance model of Zhang et al. (2003), which is very similar to the Wesely formulations with regard to $R_a$ and $R_b$. However, the $R_c$ is expressed as several more serial and parallel resistances, based on Wesely (1989) but with some

adjustments within CAMx (Ramboll Environment and Health, 2020).

EMEP deposition mechanisms are not described here, as EMEP does not deliver separate $NO_2$ and $O_3$ deposition files and will not be considered in Sect. 3.4.

## 2.5 Observational Data/Statistical Analysis/Analysis of Model Results

Model results for total surface concentrations of $NO_2$ and $O_3$ from the five CTMs are evaluated against available measurements

of the air quality monitoring network taken from the download service of Air quality of the European Environment Agency EEA (https://discomap.eea.europa.eu/map/fme/AirQualityExport.htm, 2021). $NO_2$ concentrations are monitored at 67 and $O_3$ at 53 background stations. Figure 1 shows the locations of the measurement stations, and detailed information on the stations is given in Appendix B.

The criteria for the selection of the stations were i) station type is "background", ii) elevation is below 1000 m and iii) data for

more than one of the pollutants $NO_2$, $O_3$ or $PM_{2.5}$ are available. The latter was chosen for further comparison in this intercomparison project. Preferably, stations close to the sea were chosen since modeling ship contributions were the major focus of this study. Furthermore, the domain was divided into four parts ("west", "north", "south", "east"), and a roughly equal number of stations should be in each parcel (map in Supplements Figure S1). The measured concentrations at the stations were compared against the output of the CTMs. For this purpose, the grid cell of the respective monitoring station was determined,

and modeled concentrations were taken from there.





To quantify the model performance, the root mean square error of the modeled values (RMSE), normalized mean bias (NMB) and Spearman's correlation coefficient (R) were calculated for each monitoring station, as described in Appendix A. A categorization for correlation was performed as described in Schober et al. (2018), adjusted and displayed in Table 2.

**Table 2: Interpretation of the correlation coefficient, as described in Schober et al (2018), adjusted**.

| Magnitude of Correlation Coefficient | Interpretation |
|---|---|
| **0.00–0.39** | Weak correlation |
| **0.40–0.69** | Moderate correlation |
| **0.70–1.00** | Strong correlation |

Time series were used to compare the modeled daily mean concentrations to observations at exemplary stations. In addition, the annual mean ship contribution was calculated based on hourly data. For a graphical comparison of the model performances R, NMB and RMSE, boxplots were used based on annual values calculated from hourly data at each station. For the

intercomparison maps, annual mean values based on the hourly data are used. The correlation R between models was calculated for each grid cell based on hourly data.





## 3 Results and Discussion

In the following section, the results for NO$_2$ and O$_3$ model performance and spatial distribution will be shown. Afterward, O$_x$
and NO$_x$ will be displayed for a more detailed investigation of the photochemistry and lifetime of the species. The results of
dry deposition of NO$_2$ and O$_3$ will be considered in Sect. 3.4.

### 3.1 Model Performance and Intercomparison

To evaluate the performance of the models, modeled concentrations considering all emission sectors (base case) for annual
values of 2015 were compared against actual measured data of NO$_2$ and O$_3$. Based on the results of the five models for the
cases with (base case) and without shipping emissions (noship case), contributions of the shipping sector to the NO$_2$ and O$_3$
concentrations were estimated. Maps display the annual mean values for 2015 and the relative ship contributions.
With this setup, the model performance and ship contribution of the different models can be directly compared.

### 3.1.1 NO$_2$ Model Performance

Table 2 contains R, NMB and RMSE based on the annual time series for NO$_2$ at all stations. The highest correlation across all
67 stations showed LOTOS-EUROS followed by CMAQ with a slightly lower correlation (LOTOS-EUROS: R = 0.45;
CMAQ: R = 0.43), whereas for CHIMERE, EMEP and CAMx, an overall weak correlation was found (R = 0.06 to R = 0.09).
The NMB suggests that all five CTMs underestimate the annual mean concentrations at most measurement sites; the NMB for
all stations is negative for all models. The RMSE is within the same range for all models (RMSE = 15.6 µg/m³ to 19.8 µg/m³;
Table 2).
Time series for two example stations show the temporal variations between measured and modeled data. The supplements
provide an overview of the mean values of stations in each map parcel ("west", "north", "south", "east"; Supplement Figure
S1). Figure 2 displays a time series at an urban background station in France (fr08614, "Gauzy", latitude: 43.8344, longitude:
4.374219), which was chosen because southern France will be investigated in greater detail as part of this study. Figure 3
shows a rural background station in Italy (it1773a, "Genga – Parco Gola della Rossa", latitude: 43.46806, longitude: 12.95222),
which was chosen due to its central location in the domain and the high number of stations in Italy. Figure 4 displays the time
series at a station in Greece (gr0035a, "Lykovrysi", latitude: 38.06963, longitude: 23.77689) to include a station in the eastern
part of the domain.
Measurements at the French station show the highest NO$_2$ values in winter, with peaks between 40 µg/m³ and 55 µg/m³ (Figure
2). LOTOS-EUROS and EMEP underestimate the values throughout the year. Moderate correlation was calculated for CMAQ
(R = 0.6) and LOTOS-EUROS (R = 0.65) at this station. The modeled ship contribution has annual mean values from 0.2
µg/m³ (EMEP, CAMx) to 0.6 µg/m³ (CMAQ) at station fr08614. Shipping emissions have a relative contribution between 1.8
% (EMEP) and 6.7 % (CMAQ) to the total concentration in the annual mean. The highest ship contribution at this station was
modeled by CMAQ. At the Italian station, 1773a lower NO$_2$ concentrations were measured compared to the station in France.



The highest peaks are approximately 20 µg/m³ in winter. At station it1773a, the ship contribution to the total NO₂ concentration
has annual mean values between 0.07 µg/m³ (LOTOS-EUROS) and 0.5 µg/m³ (CAMx). The highest relative ship contribution
was 7.9 % and was modeled by CAMx. At station gr0035a, the lowest simulated values are shown by CMAQ and LOTOS-
EUROS. The highest values display EMEP at this station, also with the highest correlation between measured and modeled
data (R = 0.55). The ship contribution at the Greek station is between 5.0 % (EMEP) and 15.3 % (CAMx), which is higher
than the ship contribution at the other two stations.

All models underestimate the actual measured total NO₂ values at both stations, except for LOTOS-EUROS in Italy. None of
the models are able to model matching peak values. Neither at the station in France, Italy or Greece models showed seasonal
variation in concentrations, whereas NO₂ usually has higher values in winter and lower values in summer, mainly because of
lower photolytical degradation and suppressed vertical mixing, as described, i.e., in Ordóñez (2005).

Differences in ship contributions between the stations are caused by the location and station type (fr08614 = urban background;
it1773a = rural background; gr0035a = suburban background). At the French station, the traffic-related NO₂ concentration
might supersede the ship-related NO₂. The station in Italy is not located in a city, so the NO₂ concentration caused by ships
comes to the fore. The highest ship contribution was simulated at the station in Greece because it is suburban but close to the
Port of Piraeus, which is one of the largest ports in the Mediterranean Sea. As expected, the average ship contribution is low
at stations that are not directly located at the coast or to a harbor.

To compare the correlation R, NMB and RMSE at all measurement stations for all models, the results of the comparison are
divided by country and displayed in boxplots (Figure 5). Each dot displays one measurement station. The correlation measured
against the modeled annual mean NO₂ is highest for LOTOS-EUROS and CMAQ in all countries, reflecting the results shown
in Table 3 for correlation. Nevertheless, boxplots for NMB and, in particular, for RMSE visualize that differences among
countries are larger than differences among the models (Figure 5 b, c). This means that all models show good or bad
performance at some stations, which was not found to be statistically relevant.

Underestimations by models of NO₂ at urban sites were found in other studies (Karl et al., 2019a; Giordano et al., 2015),
despite differences in grid size. Karl et al. (2019a) used a grid resolution of 4 km, and Giordano et al. (2015) used a grid
resolution of ~ 0.25° (27 km to 28 km). The underestimation might be due to too low emissions in the inventory used by the
models and the heterogeneity of emissions. Regional models cannot display small-scale spatial heterogeneity; coarse grid cells
are not representative of the measurement location. Giordano et al. (2015) suggested in their study that the underestimation of
NO₂ could be caused by either an underestimation of the chemical lifetime of NOₓ, excessively high dry deposition, an
underestimation of natural emissions at rural and remote stations or a combination of these factors. Differences in radical
concentrations and reactive nitrogen might be additional reasons for underestimation (Knote et al., 2015).

The model performance of NO₂ has shown that differences in time series between the models occur, caused by the large grid
size and the differences in meteorology.



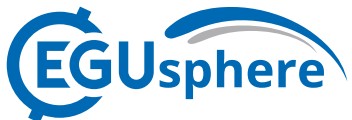

**Table 3: Correlation, normalized mean bias (NMB), root mean square error (RMSE), observational (obs) and modeled (mod) mean values of NO₂ for 2015: first data were averaged stationwise and then averaged for all 67 stations.**

|  | Correlation R | NMB | RMSE (µg/m³) | mod (µg/m³) | obs (µg/m³) |
|---|---|---|---|---|---|
| **CAMx** | 0.06 | -0.34 | 19.8 | 8.2 | |
| **CHIMERE** | 0.09 | -0.54 | 19.0 | 5.7 | |
| **CMAQ** | 0.43 | -0.56 | 17.6 | 6.9 | 17.2 |
| **EMEP** | 0.09 | -0.42 | 19.3 | 7.1 | |
| **LOTOS-EUROS** | 0.45 | -0.52 | 15.6 | 7.6 | |






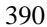

**Figure 2: Time series with daily mean NO₂ concentrations in 2015 at station fr08614 in France. The black triangle on the map (bottom right) displays the location of the station. (a) = CAMx, (b) = CHIMERE, (c) = CMAQ, (d) = EMEP, (e) = LOTOS-EUROS. Dashed grey line = measured data, colored lines = modelled data, grey line = modelled ship contribution. Correlation between modelled and measured data for hourly total emission data for 2015: CAMx = 0.23, CHIMERE = 0.20, CMAQ = 0.60, EMEP = 0.02, LOTOS-EUROS = 0.65. Ship$_a$ displays absolute ship contribution, Ship$_r$ relative ship contribution of the respective model. (t) = tagging, (z) = zero-out method for LOTOS-EUROS.**





**Figure 3: Time series with daily mean NO₂ concentration in 2015 at station it1773a in Italy. The black triangle on the map (bottom right) displays the location of the station. (a) = CAMx, (b) = CHIMERE, (c) = CMAQ, (d) = EMEP, (e) = LOTOS-EUROS. Dashed grey line = measured data, colored lines = modelled data, grey line = modelled ship contribution. Correlation between modelled and measured data for hourly total emission data for 2015: CAMx = 0.03; CHIMERE = 0.03; CMAQ = 0.20; EMEP = -0.09; LOTOS-EUROS = 0.14 Shipₐ displays absolute ship contribution, Shipᵣ relative ship contribution of the respective model. (t) = tagging, (z) = zero-out method for LOTOS-EUROS.**





**Figure 4: Time series with daily mean NO₂ concentration in 2015 at station gr0035a in Greece. The black triangle on the map (bottom right) displays the location of the station. (a) = CAMx, (b) = CHIMERE, (c) = CMAQ, (d) = EMEP, (e) = LOTOS-EUROS. Dashed grey line = measured data, colored lines = modelled data, grey line = modelled ship contribution. Correlation between modelled and measured data for hourly total emission data for 2015: CAMx = 0.15; CHIMERE = 0.20; CMAQ = 0.28; EMEP = 0.55; LOTOS-EUROS = 0.38. Ship$_a$ displays absolute ship contribution, Ship$_r$ relative ship contribution of the respective model. (t) = tagging, (z) = zero-out method for LOTOS-EUROS.**



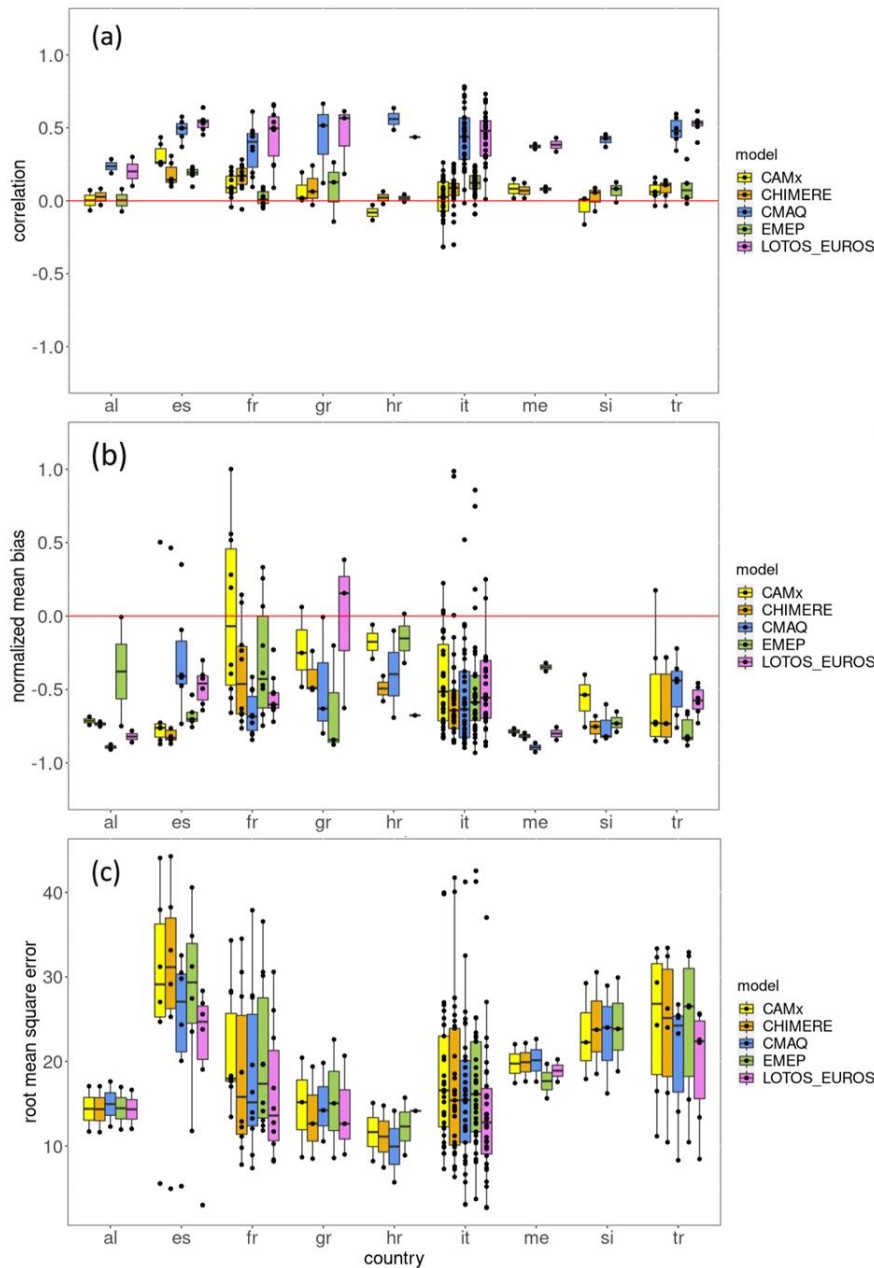

**Figure 5: (a) = Correlation, (b) = NMB, (c) = RMSE for annual mean NO₂ concentration based on hourly data. Dots display annual mean values at measurement stations for the respective countries (al= Albania; es = Spain; fr =France; gr = Greece; hr = Croatia; it = Italy; me = Montenegro; si = Slovenia; tr = Turkey). Boxplots are for the models with the boxes displaying the interquantile range (IQR) between the 25th (Q1) and 75th (Q3) percentile, the black line displays the median (Q2), whiskers are calculated as Q1–1.5*IQR (minimum) and Q3 + 1.5*IQR (maximum).**



### 3.1.2 NO₂ Spatial Distribution

The modeled annual mean $NO_2$ concentrations considering all emission sectors are similar for all models, with most values between 0 µg/m³ and 2 µg/m³ (Figure 7). CAMx and CHIMERE have the largest areas, with values exceeding 5 µg/m³, especially along the main shipping routes and in urban areas. The CMAQ, EMEP and LOTOS-EUROS maps look similar, which is in good agreement with the displayed time series in Sect. 3.1., where the results are within the same range.

Over land area, all model outputs display a concentration pattern ranging within one order of magnitude. Nevertheless, the

frequency distributions of the CMAQ, EMEP and LOTOS-EUROS model outputs show the highest frequency between 0.5 µg/m³ and 1.5 µg/m³, whereas for CAMx and CHIMERE, they are more equally distributed. Higher values of $NO_2$ concentrations simulated by CAMx and CHIMERE indicate a longer lifetime of $NO_2$ in the atmosphere. $NO_2$ reacts quickly with hydroxyl radicals (OH) and forms $HNO_3$, or $NO_2$ photolysis creates $O_3$ during the daytime. The correlation between the models for total $NO_2$ concentration was calculate based on hourly data (Table 4). The highest correlation was found between

CAMx and CHIMERE outputs (R = 0.80), but EMEP and CMAQ output were also within one range, demonstrating a strong correlation (R = 0.74). Weak correlations were found between LOTOS-EUROS and CAMx (R = 0.25) and LOTOS-EUROS and CHIMERE (R = 0.26). This weak correlation is due to the differences in frequency distribution, with LOTOS-EUROS showing most values below 1 µg/m³, whereas for CAMx and CHIMERE, more values are located in the higher value ranges. Overall, the models can give a robust estimate regarding the base run of the annual mean of $NO_2$.

The highest contribution of ships to total $NO_2$ concentrations was found at the main shipping routes, with values > 85 % (Figure 8). Similar values were found for the Baltic Sea (Karl et al., 2019a) and for the Iberian Peninsula (Nunes et al., 2020). CHIMERE and CAMx model the highest values over the sea region, expecting a ship contribution to $NO_2$ between 60 % and 85 %. CMAQ, LOTOS-EUROS and EMEP have similar patterns for ship contributions over the sea.

On the Mediterranen coastline, CMAQ, CHIMERE, LOTOS-EUROS and EMEP simulate a similar contribution, with 25 %

to 45 % ship contributions to total $NO_2$. The CAMx model reveals a higher contribution with > 85 % at the coastline. The ship contribution displayed in the time series in Sect. 3.1 was lower, although the measurement stations were not far from the coast. This shows that although the contribution from ships reaches regions far from the coast, the highest impact is over the sea area. The frequency distribution for the relative ship contribution shows that all models simulate most values between 0 % and 5 % of the ship contribution. Interestingly, the distribution is lowest at values between

20% and 40% (CMAQ, EMEP, LOTOS-EUROS) and 60 % (CAMx, CHIMERE) and then increases again at higher values, showing a bimodal distribution. This is due to large areas with high contributions over water and large areas with low contributions over land.

Over land in the northeast area of the domain, slightly negative ship contributions are derived from the CMAQ, CAMx, LOTOS-EUROS and EMEP model results. CHIMERE shows only very few negative values, but in the same region. Negative

ship contributions to $NO_2$ concentrations may arise when the zero-out method is applied. They might be a consequence of the




nonlinear NO$_x$ gas phase chemistry. Especially in areas where the impact of NO$_x$ emissions from shipping is very low, less NO oxidation takes place because the additional NO from shipping in other areas already consumed the oxidants (e.g., O$_3$). The boxplots in Figure 6 display the annual mean values for the whole model domain of NO$_2$. Model outputs vary for the base run but also for the ship contribution output. This variability needs to be taken into account when the predictive power of

models is considered. The "all_mean" boxplot displays the mean of all models and displays that in comparison with other models, CAMx has high values. It further helps to show which models tend to simulate higher or lower values compared to others. The "all_mean" boxplots show similar ranges as boxplots for CMAQ and EMEP, particularly regarding absolute and relative ship contributions. Additionally, models simulating a higher overall concentration of pollutants also tend to simulate a higher ship contribution. The relative ship contribution is highest for CAMx and CHIMERE and lowest for LOTOS-EUROS.


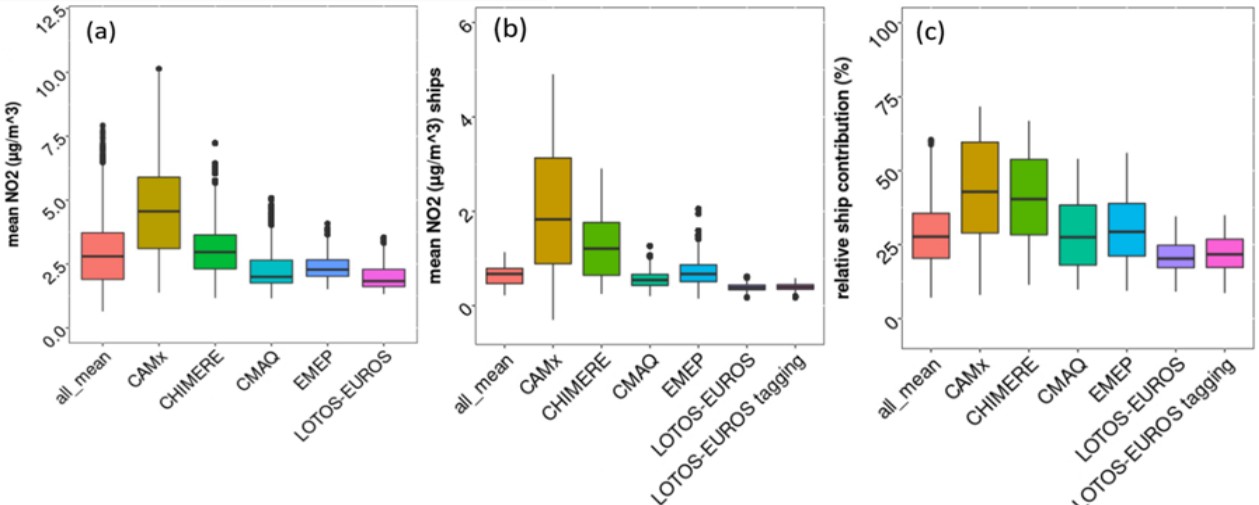

**Figure 6: Annual mean for all grid cells in the whole model domain. (a) = mean NO$_2$ for all emission sectors (base case), (b) = mean NO$_2$ for shipping only, (c) = relative ship contribution to total NO$_2$ concentration. All_mean is the mean value of all models, with a median of (a) = 2.8 μg/m³, (b) = 0.7 μg/m³ and (c) = 27.7 μg/m³.**

**Table 4: Correlation for the NO$_2$ base run between models for the whole domain (all grid cells), based on hourly data for NO$_2$ total concentration.**

| all | CAMx | CHIMERE | CMAQ | EMEP | LOTOS-EUROS |
|---|---|---|---|---|---|
| **LOTOS-EUROS** | 0.25 | 0.26 | 0.54 | 0.59 | - |
| **EMEP** | 0.40 | 0.44 | 0.74 | - | |
| **CMAQ** | 0.40 | 0.44 | - | | |
| **CHIMERE** | 0.80 | - | | | |
| **CAMx** | - | | | | |





**Figure 7: Annual mean NO₂ total concentration. (a) = CAMx, (b) = CHIMERE, (c) = CMAQ, (d) = EMEP, (e) = LOTOS-EUROS. Below the maps is the respective frequency distribution displayed for the annual mean NO₂ concentration, referred to the whole model domain.**







**Figure 8: Annual mean NO₂ ship contribution. (a) = CAMx, (b) = CHIMERE, (c) = CMAQ, (d) = EMEP, (e) = LOTOS-EUROS. Below the maps is the respective frequency distribution displayed for the annual mean NO₂ ship contribution, referred to the whole model domain.**






### 3.1.3 LOTOS-EUROS: zero-out vs. tagging

The LOTOS-EUROS model used two methods to calculate the ship contribution for $NO_2$. The range of values calculated with the zero-out method for ship contribution is larger compared to the tagging method, reaching from -2.5 % over land areas to 85 % at the main shipping lanes (Figure 9, a). By using the tagging method, ship
contributions range from 0.2 % over land areas to 75 % at the main shipping lanes (Figure 9, b). The tagging method does not produce negative values. Regarding the overall output in boxplots (Figure 6), ship contribution for both methods is within the same range.

Although all models use relatively precise higher-order algorithms for chemical calculations, they still have a certain amount of numerical noise, causing over- or underestimation of certain emission sources when using the zero-out method (European
Commission, Joint Research Centre et al., 2020; Brandt et al., 2013). The tagging method simulates the concentration for shipping as an emission source parallel with the background concentrations in the CTMs and is expected to be more accurate (Brandt et al., 2013). Thürkow et al. (2021) compared the tagging method against brute force simulations of $NO_x$ with variable emission reduction percentages to study the nonlinearity. They concluded that the sector wise reductions in emissions would overestimate the base run concentration with all sectors for NO and underestimate $NO_2$ concentrations when brute force
simulations are carried out in comparison to tagging. Nevertheless, for $NO_x$, the differences were small. Small differences in $NO_x$ ship contribution between the tagging and zero-out methods were also found in the present study (Figure 9 c, d).

However, the preference of the method that shall be used for quantifying the ship contribution also depends on the question that needs to be answered. Zero-out focuses on a situation that would appear when emissions from a certain source are shut off entirely, whereas the tagging method assigns a relative value to each source. In addition, for comparing the ship contribution
output of different models, the zero-out method is the most common way to obtain a standardized output. For the comparison of model outputs with regard to the shipping contribution, zero-out is an adequate method. Furthermore, the tagging method used in the present study only traces emission-preserved atoms (i.e., carbon or nitrogen). Thus, it did not produce a source allocation for $O_3$ in shipping emissions. Mertens et al. (2018) introduced an advanced tagging method for the contribution of land transport and shipping emissions to $O_3$, which is not yet included in LOTOS-EUROS, which can resolve the problem of
lacking ship contributions to $O_3$. A sensitivity run with stepwise reduction of $NO_x$ emissions for the zero-out method could hint at a possible shift in the atmospheric photochemical regime. However, this was not the focus of the present study.





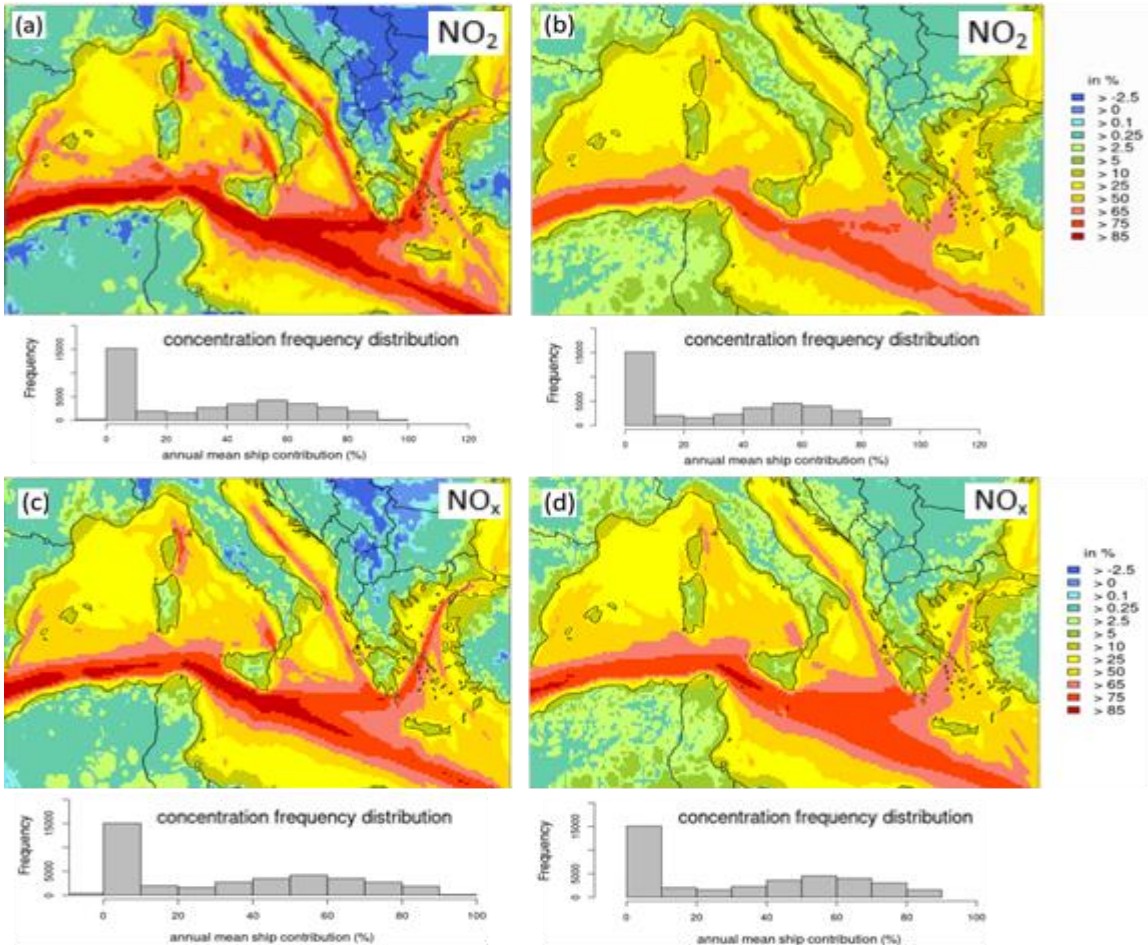

**Figure 9: Annual mean ship contribution for NO₂ (upper two maps) and NOₓ (lower two maps), calculated with LOTOS-EUROS (a) = zero-out method ship contribution NO₂, (b) = tagging method ship contribution NO₂, (c) = zero-out method ship contribution NOₓ, (d) = tagging method ship contribution NOₓ. Below the maps is the respective frequency distribution displayed for the annual mean NO₂ and NOₓ ship contribution, referred to the whole model domain.**





### 3.1.4 O$_3$ Model Performance

The tropospheric O$_3$ concentrations are strongly connected to the NO$_2$ concentration and to the oxidized nitrogen chemistry in
the atmosphere. O$_3$ can be both an initiator and a product of photochemistry; thus, it is crucial in tropospheric chemistry.
Modeled versus measured data of one-year daily mean O$_3$ time series show a weak (EMEP: R = 0.38) to moderate correlation
(CAMx: R = 0.40; CHIMERE: R = 0.48; CMAQ: R = 0.60; LOTOS-EUROS: R = 0.69; Table 5).
Selected time series represent these differences in correlation. Nevertheless, for the first months of the year CHIMERE, CAMx
and CMAQ overestimate the actual measured O$_3$ values (Figure 10: station fr08614; Figure 11: station it1773a; Figure 12:
gr0035a).
During summer months, O$_3$ shows the highest values due to increased photochemical activity. The modeled ship contribution
is between 1.1 µg/m³ (CAMx) and 2.8 µg/m³ (LOTOS-EUROS) at station fr08614 and has a relative contribution between 1.3
% (CAMx) and 4.0 % (CHIMERE) to the total concentration. At station it1773a, the mean O$_3$ ship contribution is between 1.0
µg/m³ (CAMx) and 3.0 µg/m³ (CHIMERE), and the relative contribution ranges from 1.1 % (CAMx) to 3.5 % (LOTOS-
EUROS). The ship contribution of station gr0035s ranges from -0.1 µg/m³ (CAMx) to 3.7 µg/m³ (CMAQ; LOTOS-EUROS),
which is a relative contribution of -0.1 % (CAMx) and 3.7 % (CMAQ).
The O$_3$ ship contribution is within the same range at both stations and for all five models. Figure 13 shows that CMAQ has the
smallest bias compared to the other models (NMB = 0.29), followed by LOTOS-EUROS (NMB = 0.36). The RMSE is lowest
for CMAQ (RMSE = 32.0 µg/m³) and LOTOS-EUROS (RMSE = 32.6 µg/m³), along with the lower NMB compared to the
other models. The performance analysis revealed that all five models predict higher O$_3$ concentrations than those measured at
almost all stations (NMB > 0). The overestimation of actual measured O$_3$ by the models is in line with results from previous
studies (Karl et al., 2019a; Appel et al., 2017; Im et al., 2015a; Im et al., 2015b). Im et al. (2015a) showed that O$_3$ concentrations
above 140 µg/m³ are underestimated, while concentrations below 50 µg/m³ are overestimated by 40 % to 80 % in all considered
models. This overestimation of O$_3$ by the models is likely linked to the chemical boundary conditions used in the regional
CTMs. Analyses of the boundary conditions revealed that, especially in winter, O$_3$ levels are mostly driven by transport instead
of local production due to limited photochemistry (Giordano et al., 2015).
CHIMERE uses boundary conditions from monthly mean climatologies simulated with the LMDz-INCA model, CAMx uses
Mozart-4 output, LOTOS-EUROS and CMAQ use IFS-CAMS reanalysis data and the EMEP model uses ozone boundary
conditions provided with the open source model distribution for 2015. These differences in input for the boundary conditions
can be seen as the reason for the varying output in O$_3$.
All models performed relatively well and are able to represent the course of the year, with higher values in summer and lower
values in winter. Nevertheless, in some cases, the values in spring are overestimated.





**Table 5: Correlation, normalized mean bias (NMB), root mean square error (RMSE), observational (obs) and modeled (mod) of O$_3$**
**as the mean values for 2015: the first data were averaged stationwise and then averaged for all 53 stations.**

|  | Correlation R | NMB | RMSE (µg/m³) | mod (µg/m³) | obs (µg/m³) |
|---|---|---|---|---|---|
| **CAMx** | 0.40 | 0.45 | 41.8 | 90.6 | |
| **CHIMERE** | 0.48 | 0.62 | 47.0 | 101.2 | |
| **CMAQ** | 0.60 | 0.29 | 32.0 | 81.2 | 65.2 |
| **EMEP** | 0.38 | 0.42 | 40.4 | 87.9 | |
| **LOTOS-EUROS** | 0.69 | 0.36 | 32.6 | 87.7 | |



**Figure 10: Time series with daily mean O₃ concentration in 2015 at station fr08614 in France. The black triangle on the map (bottom right) displays the location of the station. (a) = CAMx, (b) = CHIMERE, (c) = CMAQ, (d) = EMEP, (e) = LOTOS-EUROS. Dashed gray line = measured data, colored lines = modeled data, gray line = modeled ship contribution. Correlation between modeled and measured data for hourly total emission data for 2015: CAMx= 0.57; CHIMERE = 0.6; CMAQ = 0.71; EMEP = 0.39; LOTOS-EUROS = 0.78. Ship$_a$ displays absolute ship contribution, Ship$_r$ relative ship contribution of the respective model.**





**Figure 11: Time series with daily mean O₃ concentration in 2015 at station it1773a in Italy. The black triangle on the map (bottom right) displays the location of the station. (a) = CAMx, (b) = CHIMERE, (c) = CMAQ, (d) = EMEP, (e) = LOTOS-EUROS. Dashed gray line = measured data, colored lines = modeled data, gray line = modeled ship contribution. Correlation between modeled and measured data for hourly total emission data for 2015: CAMx = 0.37; CHIMERE = 0.4; CMAQ = 0.58; EMEP = 0.35; LOTOS-EUROS = 0.7. Ship_a displays absolute ship contribution, Ship_r relative ship contribution of the respective model.**








**Figure 12: Time series with daily mean O₃ concentration in 2015 at station gr0035a in Greece. The black triangle on the map (bottom right) displays the location of the station. (a) = CAMx, (b) = CHIMERE, (c) = CMAQ, (d) = EMEP, (e) = LOTOS-EUROS. Dashed gray line = measured data, colored lines = modeled data, gray line = modeled ship contribution. Correlation between modeled and measured data for hourly total emission data for 2015: CAMx = 0.29; CHIMERE = 0.46; CMAQ = 0.50; EMEP = 0.71; LOTOS-EUROS = 0.57. Ship$_a$ displays absolute ship contribution, Ship$_r$ relative ship contribution of the respective model.**




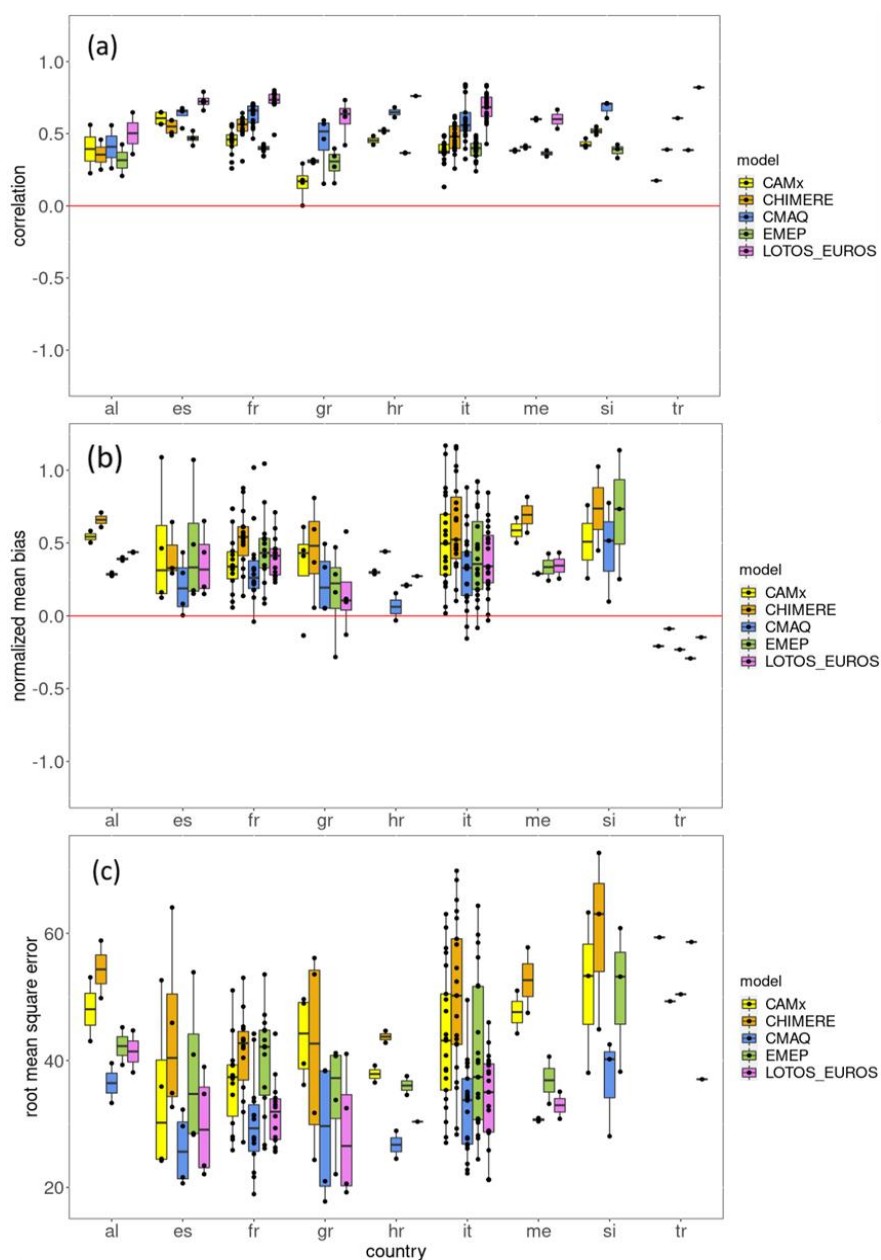

**Figure 13: (a) = Correlation, (b) = NMB, (c) = RMSE for annual mean O₃ concentration. Dots display values at measurement stations for the respective countries (al= Albania; es = Spain; fr =France; gr = Greece; hr = Croatia; it = Italy; me = Montenegro; si = Slovenia; tr = Turkey). Boxplots are for the models with the boxes displaying the interquantile range (IQR) between the 25th (Q1) and 75th (Q3) percentile, the black line displays the median (Q2), whiskers are calculated as Q1–1.5*IQR (minimum) and Q3 + 1.5*IQR (maximum).**



### 3.1.5 O$_3$ Spatial Distribution

The annual mean concentration of O$_3$ considering all emission sectors is between 60 µg/m³ and 120 µg/m³ for all models
(Figure 15). This is consistent with the measurements displayed in the time series in Sect. 3.2.1. CHIMERE, CAMx and
LOTOS-EUROS show particularly high O$_3$ concentrations over the sea. Interestingly, EMEP results are similarly high over
the sea area, but in comparison with other models, concentrations are lower over land, and even values below 60 µg/m³ can be
seen in the Po valley (Figure 15, d). Regarding the correlation between the models for total concentration over the whole
domain, it is highest between CMAQ and EMEP (R = 0.71) and lowest for CAMx and LOTOS-EUROS (R = 0.39), but
predominantly moderate correlations were found among the models (Table 6).

In general, all models show high annual mean concentrations over the sea areas and low annual mean concentrations over land
areas, which might be traced back to the emission input datasets that were split into land-based emissions and emissions from
oceangoing ships. Furthermore, high values of O$_3$ are expected to enter the domain from the eastern part of the Mediterranean
Sea. This point will be discussed in Sect. 4. The frequency distribution of the annual mean total concentration of O$_3$ has a
bimodal distribution for CHIMERE, CMAQ and EMEP. This reflects photochemical O$_3$ depletion or production, with high
values over water areas and lower values over land. Over water, low O$_3$ depletion is expected during the night. A comparison
of diurnal cycles of O$_3$ over water and over land shows that this presumption is reflected by CMAQ and EMEP output, showing
more pronounced cycles of O$_3$ in grid cells over land (Appendix C). However, the diurnal cycles of CAMx, CHIMERE and
LOTOS-EUROS do not show differences in amplitude over land and water. Despite this, over water, all models show a higher
spread of values within diurnal cycles, displaying that there is more variability in the course of the year over water than over
land.

The relative contribution of ships to total O$_3$ concentrations is lowest in areas with a high contribution of shipping to total NO$_2$
(Figure 16). It decreases to -20 % in areas with high NO$_2$ concentrations in all model outputs, displaying a local scale titration
of O$_3$ by NO, which is emitted by ships. This reverse relationship between NO$_2$ and O$_3$ was already shown in other studies
(e.g., Karl et al., 2019b). Consequently, the largest areas with O$_3$ destruction for the CAMx and CHIMERE models coincide
with areas where the models show the highest contribution of shipping to NO$_2$. The comparison with the time series shows the
highest ship contribution to the total O$_3$ concentration in summer. Likewise, in Sect. 3.1.4 lowest ship contribution was found
for CAMx.

Figure 14 shows boxplots with annual mean values of the models for the whole domain. It shows that CAMx, CHIMERE and
LOTOS-EUROS are within one range regarding the annual mean total concentration. The CMAQ and EMEP outputs are
lowest for the annual mean O$_3$ total concentration. Regarding ship contribution, all models except CAMx are within one range.
The present study does not contain the parts of the Mediterranean Sea furthest east due to the focus of the project on the western
Mediterranean Sea with its harbor cities as well as due to the limited extent of the WRF domain. A more detailed investigation
of the boundary conditions of CMAQ has shown high O$_3$ values in the eastern part of the domain. A high O$_3$ production over
the eastern Mediterranean Sea and a steep west-east gradient of O$_3$ were described in previous studies (i.e., Doche et al., 2014;



Safieddine et al., 2014; Liu et al., 2009). This production influences the amount of $O_3$ in the western part of the Mediterranean Sea. Safieddine et al. (2014) found an increase of up to 22 % in $O_3$ in the eastern part of the Mediterranean basin compared to the middle of the basin. Doche et al. (2014) described a steep west–east $O_3$ gradient with the highest concentrations over the eastern part of the Mediterranean basin.

Overall, all models showed a relatively good performance for $O_3$ but differed in modeling spatial distribution and ship contribution mainly over water. Although boxplots for annual mean values of $O_3$ differ, for relative ship contribution they show that CHIMERE, CMAQ, EMEP and LOTOS-EUROS are within one range. Diurnal cycles did not reveal differences in $O_3$ depletion over water and land between the models.

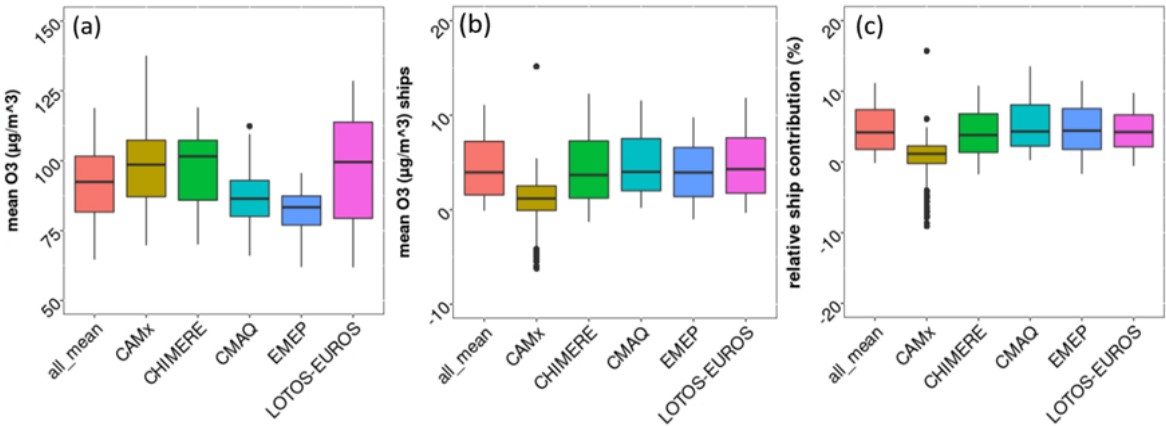

**Figure 14: Annual mean for the whole model domain. (a) = mean $O_3$ for all emission sectors (base case), (b) = mean $O_3$ for shipping only, (c) = relative ship contribution to total $O_3$ concentration. All_mean is the mean value of all models, with a median of (a) = 92.4 µg/m³, (b) = 4.0 µg/m³ and (c) = 4.2 µg/m³.**

**Table 6: Correlation between models for the whole domain (all grid cells) based on hourly data for $O_3$ total concentration.**

| all | CAMx | CHIMERE | CMAQ | EMEP | LOTOS-EUROS |
|---|---|---|---|---|---|
| **LOTOS-EUROS** | 0.39 | 0.56 | 0.49 | 0.55 | - |
| **EMEP** | 0.44 | 0.58 | 0.71 | - | |
| **CMAQ** | 0.50 | 0.56 | - | | |
| **CHIMERE** | 0.63 | - | | | |
| **CAMx** | - | | | | |





**Figure 15: Annual mean O₃ total concentration. (a) = CAMx, (b) = CHIMERE, (c) = CMAQ, (d) = EMEP, (e) = LOTOS-EUROS; emisbase maps, annual mean value, white areas contain values below 60 µg/m³. Below the maps is the respective frequency distribution displayed for the annual mean O₃ concentration, referred to the whole model domain.**





**Figure 16: Annual mean O₃ ship contribution. (a) = CAMx, (b) = CHIMERE, (c) = CMAQ, (d) = EMEP, (e) = LOTOS-EUROS; white areas display values below -20 %. Below the maps is the respective frequency distribution displayed for the annual mean O₃ ship contribution, referred to the whole model domain.**





### 3.2 $O_x$ Spatial Distribution

The oxidation of VOCs produce $O_3$ in the troposphere when nitrogen oxides (NO; $NO_2$) and sunlight are present. Central to
understanding this production is the photostationary state formed between NO, $NO_2$, and $O_3$ in sunlight. In emission-free air,
a steady equilibrium would be expected; nevertheless, emission sources disturb this equilibrium. In areas with high NO
emissions, $O_3$ destruction is expected, resulting in lower $O_3$ concentrations along the main shipping routes, in urban areas and
in harbor cities.

The results show that all five models tend to underestimate $NO_2$ and overestimate $O_3$, but at different magnitudes. For a better
understanding of photochemical air pollution and chemical coupling, the oxidant levels ($O_x = O_3 + NO_2$) were calculated and
displayed for all emission sources and for the ship contribution. Clapp and Jenkin (2001) showed that the concentration of $O_x$
levels can be described as a $NO_x$-independent regional contribution, where the $O_x$ contribution equates to the $O_3$ background,
and a $NO_x$-dependent local contribution. The $NO_x$-dependent contribution correlates with the primary pollution, coming from
direct $NO_2$ emissions or VOC, which promote conversion from NO to $NO_2$ (Clapp and Jenkin, 2001).

In comparison with the $O_3$ spatial distribution and frequency distribution, the annual mean concentration of $O_x$ displays a
similar pattern for the model outputs (Figure 17). As was the case for $O_3$, the CHIMERE and CAMx models show the highest
values over the sea area, and EMEP shows the lowest values over land areas. The frequency distribution shows bimodal
distributed values for CHIMERE, CMAQ and EMEP, as it was for $O_3$. Thus, $O_x$ levels are mainly $NO_x$-independent.

Nevertheless, $NO_x$-dependent $O_x$ formation can also be seen in the ship contribution to the total $O_x$ concentration (Figure 18).
High $O_x$ contributions at the main shipping routes for CHIMERE, CMAQ, EMEP and LOTOS-EUROS indicate the local
contribution from shipping emissions ($NO_2$ and VOC), which cause high $O_x$ levels in these areas. For CAMx, such a pattern
was not found.

### 3.3 $NO_x$ Spatial Distribution

To gain further insight into the differences in the lifetime of $NO_2$ in the models, $NO_x$ (= NO + $NO_2$) was calculated and
displayed (Appendix D). Differences in $NO_x$ give a hint on the lifetimes because of the reaction of $NO_2$ with OH to $HNO_3$.
The latter forms ammonium nitrate aerosol together with ammonia; thus, $NO_2$ is no longer in the gaseous phase. Another
explanation is the dry deposition of $NO_2$, which also causes a loss and consequently differences in the $NO_x$ pattern due to
different                                                deposition                                                mechanisms.

The spatial distribution of the annual mean $NO_x$ and ship contribution to the total $NO_x$ concentration have shown a very similar
pattern as for $NO_2$. The values of CAMx and CHIMERE output are within one range, displaying higher values compared to
CMAQ, EMEP and LOTOS-EUROS. These three models show an output that is also within one range.

To see the chemical fate of $NO_2$ the dry deposition could give a hint and will be considered in the following Sect 3.4.




**Figure 17: Annual mean $O_x$ (= $NO_2$ + $O_3$) concentration. (a) = CAMx, (b) = CHIMERE, (c) = CMAQ, (d) = EMEP, (e) = LOTOS-EUROS. Below the maps is the respective frequency distribution displayed for the annual mean $O_x$ concentration, referred to the whole model domain.**



**Figure 18: Annual mean $O_x$ (= $NO_2$ + $O_3$) ship contribution. (a) = CAMx, (b) = CHIMERE, (c) = CMAQ, (d) = EMEP, (e) = LOTOS-EUROS. Below the maps is the respective frequency distribution is displayed for the annual mean $O_x$ ship contribution, referred to the whole model domain.**





### 3.4 Dry Deposition

In the present study, dry deposition of $NO_2$ and $O_3$ are displayed for the base and the no ship case for CAMx, CHIMERE,
CMAQ and LOTOS-EUROS. EMEP does not deliver separate $NO_2$ and $O_3$ deposition files but does deliver oxidized and
reactive nitrogen. Thus, EMEP output is not considered in this chapter.

### 3.4.1 Dry Deposition of $NO_2$

The annual mean $NO_2$ dry deposition of all four compared models displays similar values over land areas (Figure 19). In cities
and densely populated regions, all models show high $NO_2$ dry deposition, with values over 300 mg/m²/year. Nevertheless, the
frequency distribution of all values shows that this is mainly the case for CAMx and LOTOS-EUROS. Additionally, over the
sea, the pattern of annual mean dry deposition of $NO_2$ is also similar for CAMx and LOTOS-EUROS.

Table 7 shows that the correlation was strongest between CHIMERE and CAMx (R = 0.78). Similarities and strong correlations
in the output of both models were also found for the $NO_2$ concentration in Section 3.1.2. This can be traced back to the same
meteorology data that were used by both models.
The relative ship contribution to the annual dry deposition of $NO_2$ is displayed in Figure 20.
The lowest ship contribution to $NO_2$ dry deposition is modeled by CMAQ and LOTOS-EUROS. In particular, CMAQ shows
large areas with negative (-2.5 %) ship contributions over land. The CHIMERE output looks similar to the CAMx output over
land. Along the coastline, CMAQ and LOTOS-EUROS show a ship contribution between 10 % and 25 %; CAMx and
CHIMERE expect a ship contribution to the total annual deposition of 25 % to 75 %. The highest contribution is displayed by
CAMx.

Differences in $NO_2$ dry deposition model output can be due to the dry deposition velocities but also due to the different
meteorology data used by the models (Wichink Kruit et al., 2014).

Overall, the models have more differences in $NO_2$ dry deposition than in air concentration. As was the case for $NO_2$
concentration, CAMx simulated the highest values in dry deposition. The lowest values in $NO_2$ dry deposition are displayed
by CMAQ. In addition, the correlation between CMAQ and the other models was lowest.

High $NO_2$ deposition over water areas caused by ships contributes to eutrophication (Vivanco et al., 2018). A study by Im et
al. (2013) showed values of approximately 500 kg (N) m$^{-2}$ per year ($\triangleq$ 50000 mg/m²/year) over the Mediterranean Sea, which
means an exceedance of the critical load of 2 g to 3 g (N) m$^{-2}$ per year ($\triangleq$ 2000 to 3000 mg/m²/year) to marine and coastal
habitats (Bobbink and Hettelingh, 2011). The present study focused on $NO_2$ dry deposition; thus, a direct comparison with
critical load levels or with other studies regarding total N deposition would not be possible. A subsequent calculation of N
showed that the simulated values in the present study do not exceed the critical loads (Appendix E). Nevertheless, $NO_2$ dry
deposition from ships contributes to the total N deposition budget, thus increasing with ship traffic and impacting the
ecosystems in the Mediterranean Sea.



**Table 7: Correlation between models for the whole domain (all grid cells) based on hourly data for NO₂ total dry deposition.**

| all | CAMx | CHIMERE | CMAQ | LOTOS-EUROS |
|---|---|---|---|---|
| **LOTOS-EUROS** | 0.64 | 0.72 | 0.20 | - |
| **CMAQ** | 0.11 | 0.14 | - | |
| **CHIMERE** | 0.78 | - | | |
| **CAMx** | - | | | |

**Figure 19: Annual total dry deposition of NO₂. (a) = CAMx, (b) = CHIMERE, (c) = CMAQ, (d) = LOTOS-EUROS. Below the maps are the respective frequency distribution displayed for the annual mean NO₂ dry deposition, referred to the whole model domain.**





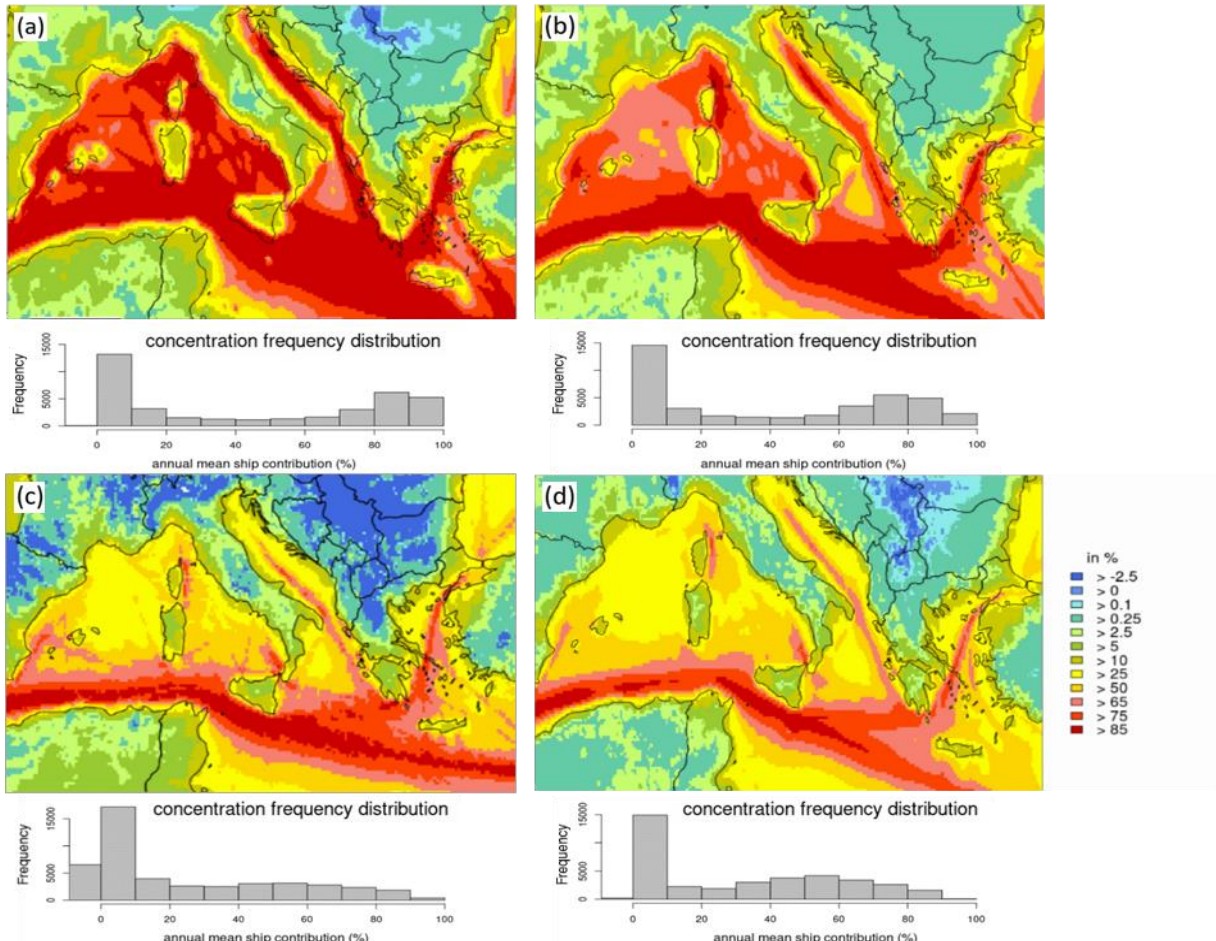

**Figure 20: Annual mean dry deposition of NO₂ relative ship contribution. (a) = CAMx, (b) = CHIMERE, (c) = CMAQ, (d) = LOTOS-EUROS. Below the maps are the respective frequency distribution displayed for the annual mean NO₂ dry deposition ship contribution, referred to the whole model domain.**





### 3.4.2 Dry Deposition O₃

Dry deposition is a major sink for $O_3$ in the lowest model layer. $O_3$ has high destruction rates on vegetated surfaces through plant stomata and lower rates on surfaces such as water or snow (Clifton et al., 2020). Spatial patterns of annual total $O_3$ dry deposition maps confirm this distribution. Over sea annual totals are lower (250 mg/m²/year to 1000 mg/m²/year) compared to values over land (2500 mg/m²/year to 10000 mg/m²/year; Figure 21). The correlation for the annual total concentration of $O_3$ dry deposition is highest between CHIMERE and CAMx, showing a moderate correlation (R = 0.59; Table 8). Nevertheless, the correlation is weak between all other models.

Figure 22 shows the ship contribution to the total dry deposition of $O_3$. CMAQ and LOTOS-EUROS are within a similar range, with ship contributions of 5 % to 10 % over water surfaces. The lowest contribution of -5 % at the main shipping lanes is modeled by CAMx, showing a similar pattern as for the $O_3$ ship contribution. Over land areas, ships contribute to dry $O_3$ deposition from 0.25 % to 2.5 %.

In addition to the impact of $O_3$ dry deposition on plant stomata, it is important to explain differences in surface $O_3$ concentration model outputs. The $O_3$ concentration is sensitive to the deposition velocity (Clifton et al., 2020), which differs among the four models. This can be confirmed by studies comparing deposition schemes, where differences in $O_3$ concentration between models are caused by the variety of processes (Clifton et al., 2020). In particular, the variability in deposition velocities across models, as discussed in Sect. 3.3.1, is seen as an originator leading to uncertainties in tropospheric $O_3$ (Wild, 2007).

A model comparison study with 15 models by Hardacre et al. (2015) found the greatest differences in total $O_3$ dry deposition occurring in areas where deposition velocities and $O_3$ concentrations are highest. Additionally, soil moisture has an important impact on $O_3$ deposition and concentration. An evaluation study within the CHIMERE model found that especially in southern Europe, where soil is close to the wilting point during summer and affects stomatal opening, $O_3$ dry deposition declines (Anav et al., 2018). This in turn affects the concentration of gases in the lower atmosphere and thus has an impact on $O_3$ concentrations.



**Table 8: Correlation between models for the whole domain (all grid cells) based on hourly data for O$_3$ total dry deposition.**

| all | CAMx | CHIMERE | CMAQ | LOTOS-EUROS |
|---|---|---|---|---|
| **LOTOS-EUROS** | 0.32 | 0.39 | 0.08 | - |
| **CMAQ** | 0.11 | 0.04 | - | |
| **CHIMERE** | 0.59 | - | | |
| **CAMx** | - | | | |





**Figure 21: Annual total dry deposition of O₃. (a) = CAMx, (b) = CHIMERE, (c) = CMAQ, (d) = LOTOS-EUROS. Below the maps are the respective frequency distribution displayed for the annual mean O₃ dry deposition, referred to the whole model domain.**



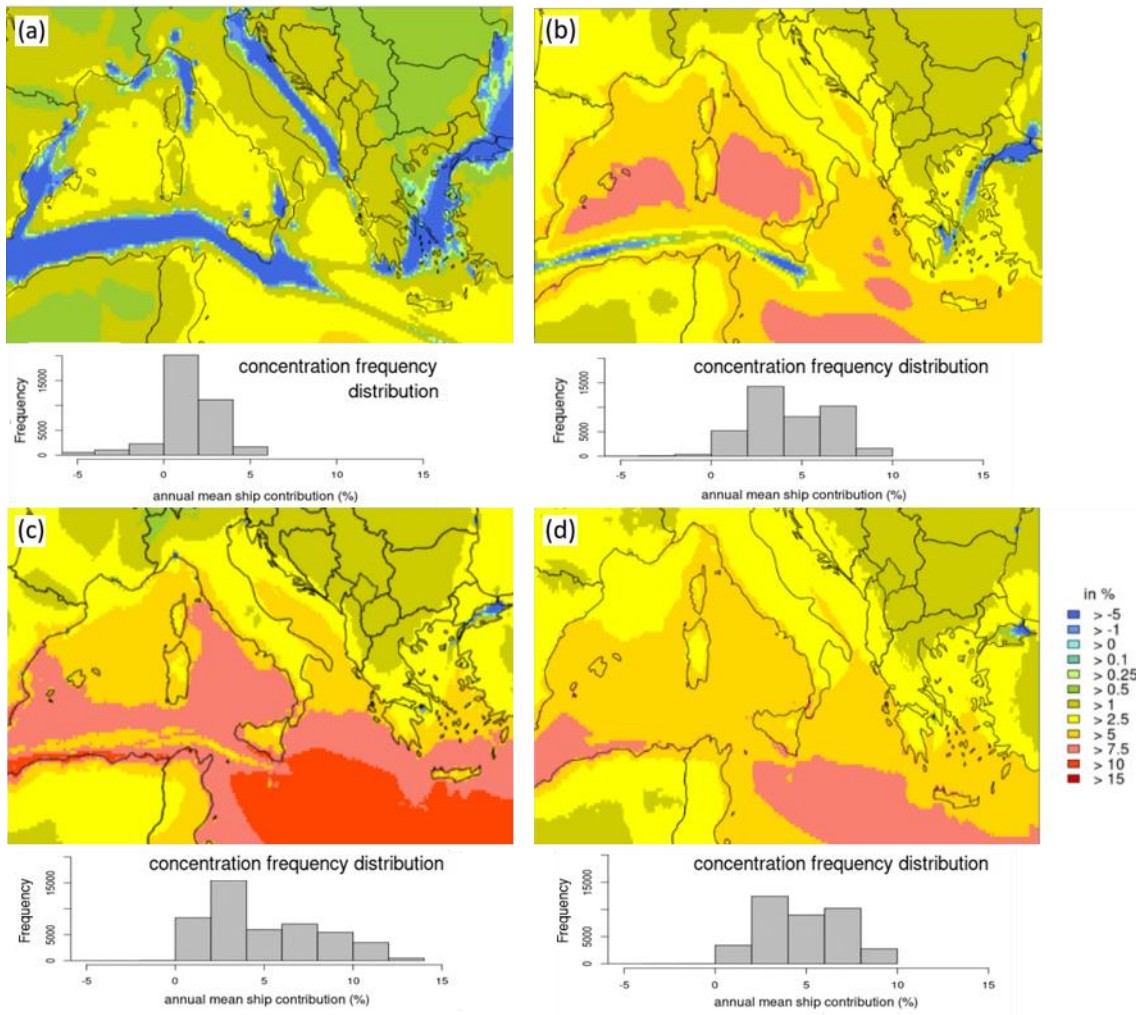

**Figure 22: Annual mean dry deposition of O₃ relative ship contribution. (a) = CAMx, (b) = CHIMERE, (c) = CMAQ, (d) = LOTOS-EUROS. Below the maps are the respective frequency distribution displayed for the annual mean O₃ dry deposition ship contribution, referred to the whole model domain.**





## 4 Summary and Conclusion


The ship contribution to air pollution in the Mediterranean Sea was simulated with five different regional-scale CTMs (CAMx, CHIMERE, CMAQ, EMEP, LOTOS-EUROS). An evaluation of the results for $NO_2$ and $O_3$ concentrations is presented here. By using different CTMs, a more robust estimate of the ship contribution to atmospheric concentrations and deposition can be obtained compared to single model runs.

The emission data, modeled year and domain were the same for all models. The models were run in their standard setup. The outputs of the model runs were quantified by comparing the measurements from urban and rural background stations around the Mediterranean Sea.

The focus of the study was the comparison of model outputs concerning the concentration of regulatory pollutants and the calculation of ship contributions to total air pollution concentrations.

Concerning the results of $NO_2$, the model performance showed differences in the time series between the models, caused by the large grid size and the differences in meteorology. All five CTMs underestimated the actual measured $NO_2$ concentration data at most stations, along with results from previous studies (e.g., Karl et al., 2019a; Giordano et al., 2015; Knote et al., 2015). The ship contribution to the total concentration of $NO_2$ at the measurement stations over land differed among the models. It was between 1.0 % and 15.3 % at the presented stations. Ship contributions mean values of several stations in one area, as

shown in the supplements Figures S2-S9, display values up to 48.1 %. This was found in the eastern part of the domain (Figure S6), where the main shipping routes are close to the shore. Studies regarding the North and Baltic Seas found similar results because shipping lanes are located closer to the shore and have a higher contribution to the total $NO_2$ concentration in coastal regions (Matthias et al., 2010; Karl et al., 2019a). Nevertheless, over water, the maps in the present study display a ship contribution > 85 % at the main shipping routes. High values are also expected for the African coast since the main shipping

route is close, but measurement stations are in continental Europe; no measurements were available for Northern Africa.

The variability in modeling the ship contribution was similar to that for the annual mean concentration of $NO_2$. In both cases, CAMx and CHIMERE displayed the highest annual mean concentration and highest relative ship contribution. CMAQ, EMEP and LOTOS-EUROS simulated values within one range, which can be confirmed by similarities in frequency distribution.

Comparison of the LOTOS-EUROS zero-out and tagging methods for $NO_2$ shows that the zero-out method models a larger

range of values for ship contribution (-2.5 % to 85 %) compared to the tagging method (0.2 % to 75 %) with the largest deviations at the main shipping lanes. The comparison of both methods for ship contributions at measurement stations displayed even smaller differences, with the highest deviation of 3.1 % in ship contributions. This leads to the conclusion that the tagging results do not largely deviate from the zero-out method.

A relatively good model performance for $O_3$ was shown by all five models, but the model outputs differed in spatial distribution

and ship contribution over water. An overestimation of $O_3$ was found at almost all stations. The overestimation of actual measured $O_3$ by models agrees with results found in other studies (Appel et al., 2017; Im et al., 2015a, b). Although boxplots for annual mean values of $O_3$ vary, for relative ship contribution they show that CHIMERE, CMAQ, EMEP and LOTOS-





EUROS are within one range. The relative contribution of ships to total O₃ decreases to -20 % in areas with high NO₂ concentrations in all model outputs, but mostly for CAMx. Diurnal cycles did not reveal differences in O₃ depletion over water

and land between the models.

The focus of the second part of the present study was dry deposition of NO₂ and O₃. The motivation to examine the dry deposition of NO₂ and O₃ more closely was to explain the model differences found for O₃ and NO₂. Investigations of dry deposition are crucial to explain the conservation of mass and fate of these substances. Although dry deposition has effects on ecosystems and human health, the impact was not a major focus of the study.

Regarding air concentration, for NO₂ dry deposition and the ship contribution, CAMx showed the highest values. CMAQ displayed the lowest values in NO₂ dry deposition. Additionally, the correlation between CMAQ and the other models was lowest.

Along the shoreline, CMAQ and LOTOS-EUROS reveal a ship contribution between 10 % and 25 %; CAMx and CHIMERE expect a ship contribution to total annual NO₂ dry deposition of 25 % to 75 %, in some regions also along the coast. These

differences are caused by mechanisms to calculate dry deposition velocities, which are unique for each model, as well as differing inputs, such as land use data (Wichink Kruit et al. 2014; Vivanco et al. 2018).

The ship contribution to the total dry deposition of O₃ displays the highest contribution with values between 75 % and 85 % by CHIMERE. CMAQ and LOTOS-EUROS are within a similar range, with ship contributions mainly 5 % to 10 % over water areas. The lowest contribution of -5 % at main shipping lanes is modelled by CAMx. The correlation of model-observation

data for the annual total concentration of O₃ dry deposition was highest for CHIMERE and CAMx. Nevertheless, no or a low correlation was found for all other models.

In general, more deviations between the dry deposition model outputs were found compared to the model outputs of the air concentration of pollutants. This is because NO₂ and O₃ in the atmosphere are formed more or less "directly" from the emission data, but dry deposition differs because there are other, model-specific mechanisms behind it.

In an additional investigation of ship contributions to air pollution, aerosol particles and wet deposition also need to be considered, which is a next step in the current intercomparison study. The aerosol formation mechanism differs in most models; therefore, a detailed investigation of PM₂.₅ and its chemical composition is necessary and will be part of further investigations in this project.

A more reliable estimate of ship contributions to the atmospheric concentration as well as deposition could be acquired when

using five different CTMs than when using only one model. This estimate can be achieved using a mean value with standard deviations of model outputs, regarding all emissions but also ship contributions, as was done in the present study. This gives a data range that is more robust and reliable compared to the output of one single model. Furthermore, possible limitations, over- and underestimations of model outputs are pointed out with the intercomparison.



**Acknowledgement**

This was work supported by SCIPPER project, which has received funding from the European Union's Horizon 2020 research and innovation programme under grant agreement Nr.814893.

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



**Appendix**

**Appendix A: Definitions of NMB, R and RMSE**

Normalized Mean Bias (NMB) $= \dfrac{\sum_1^n (M - O)}{\sum_1^n (O)}$  (1)

where $M$ and $O$ stand for model and observation results, respectively. The time average is indicated over n time intervals
(number of observations). The time average is done for one year.

Correlation (R) $= \dfrac{1}{(n-1)} \sum_1^n \left( \left( \dfrac{O - \overline{O}}{\sigma_o} \right) * \left( \dfrac{M - \overline{M}}{\sigma_m} \right) \right)$  (2)

Root Mean Square Error (RMSE) $= \sqrt{\dfrac{\sum_1^n (M - O)^2}{n}}$  (3)

RMSE is a measure of accuracy and allows prediction errors of different models to be compared for a particular dataset.





**Appendix B:**

**Table B1: detailed overview of monitoring stations**

| Name | Code | Country | Latitude | Longitude | Ele-vation | Station Type | Data Points | Measured Pollutants |
|---|---|---|---|---|---|---|---|---|
| **Vlora** | al0204a | Albania | 40.40309 | 19.4862 | 25 | urban background | 6850 | benzene, CO, $NO_2$, $NO_x$, $O_3$, $PM_{10}$, $PM_{2.5}$, $SO_2$ |
| **Shkoder** | al0206a | Albania | 42.3139 | 19.52342 | 13 | urban background | 7536 | CO, $NO_2$, $NO_x$, $O_3$, $PM_{10}$, $PM_{2.5}$, $SO_2$ |
| **Els Torms** | es0014r | Spain | 41.39389 | 0.73472 | 470 | rural background | 8549 | NO, $NO_2$, $NO_x$, $O_3$, $SO_2$ |
| **Vila-seca (RENFE)** | es1117a | Spain | 41.11209 | 1.151824 | 41 | suburban background | 8594 | NO, $NO_2$, $NO_x$ |
| **Sant Celoni (Carles Damm)** | es1275a | Spain | 41.68905 | 2.495747 | 145 | suburban background | 7180 | NO, $NO_x$, $NO_2$, $SO_2$ |
| **Barcelona (Ciutadella)** | es1679a | Spain | 41.38641 | 2.187417 | 7 | urban background | 8565 | NO, $NO_2$, $NO_x$ |
| **Mataró (passeig dels Molins)** | es1816a | Spain | 41.54716 | 2.443254 | 40 | urban background | 8484 | NO, $NO_x$, $NO_2$, $O_3$, CO |
| **Barcelona (Palau Reial)** | es1992a | Spain | 41.38748 | 2.11515 | 81 | urban background | 8393 | NO, $NO_2$, $NO_x$, $SO_2$, CO |
| **Marseille 5 Avenues** | fr03043 | France | 43.30607 | 5.395794 | 73 | urban background | 8585 | $NO_2$, $O_3$, $PM_{10}$, $PM_{2.5}$, $SO_2$ |
| **Esterel** | fr03070 | France | 43.43786 | 6.768366 | 5 | suburban background | 1820 | $NO_2$, $O_3$ |
| **Agathois-piscénois** | fr08022 | France | 43.28776 | 3.504831 | 20 | suburban background | 8382 | $NO_2$, $O_3$ |
| **Gauzy** | fr08614 | France | 43.8344 | 4.374219 | 40 | urban background | 8406 | $NO_2$, $O_3$, $PM_{10}$, $PM_{2.5}$ |
| **Rigaud** | fr08713 | France | 42.68402 | 2.903453 | 50 | urban background | 8419 | $NO_2$, $PM_{10}$ |





| | | | | | | | | |
|---|---|---|---|---|---|---|---|---|
| **Cannes Broussilles** | fr24009 | France | 43.5625 | 7.007222 | 71 | urban background | 8587 | $NO_2$, $O_3$, $PM_{10}$, $PM_{2.5}$ |
| **Manosque** | fr24018 | France | 43.83527 | 5.785831 | 385 | urban background | 8517 | $NO_2$, $O_3$, $PM_{10}$, $PM_{2.5}$ |
| **Nice Arson** | fr24036 | France | 43.70207 | 7.286264 | 11 | urban background | 8701 | $NO_2$, $O_3$, $PM_{10}$, $PM_{2.5}$ |
| **Ajaccio Sposata** | fr41007 | France | 41.94923 | 8.757586 | 60 | suburban background | 8497 | $NO_2$, $O_3$ |
| **Bastia Montesoro** | fr41017 | France | 42.67134 | 9.434644 | 47 | rural background | 8626 | $NO_2$, $O_3$, $PM_{2.5}$ |
| **Lykovrysi** | gr0035a | Greece | 38.06963 | 23.77689 | 210 | suburban background | 6719 | $NO_2$, $NO_2$, $O_3$ |
| **Neochorouda** | gr0045a | Greece | 40.73984 | 22.87623 | 229 | suburban background | 8725 | $NO_2$, $NO$, $O_3$ |
| **Finokalia** | gr0002r | Greece | 35.315871 | 25.666216 | 250 | rural background | 6825 | $PM_{10}$, $O_3$ |
| **NA** | hr0012a | Croatia | 46.16906 | 15.66064 | 0 | rural background | 6063 | $NO_2$, $NO_x$, $O_3$, $PM_{10}$, $PM_{2.5}$ |
| **NA** | hr0025a | Croatia | 44.86247 | 13.81686 | 0 | suburban background | 8293 | $NO_2$, $NO_x$, $O_3$ |
| **Melilli** | it0611a | Italy | 37.18237 | 15.12883 | 300 | urban background | 7964 | $NO_2$, $O_3$, $SO_2$ |
| **Priolo** | it0614a | Italy | 37.15612 | 15.19087 | 35 | urban background | 7902 | $NO_2$, benzene, $SO_2$ |
| **SR - Via Gela** | it0620a | Italy | 37.10247 | 15.26564 | 60 | suburban background | 6958 | $NO_2$, $O_3$, $SO_2$ |
| **Gambara** | it0741a | Italy | 45.24889 | 10.29944 | 51 | suburban background | 8413 | $NO_2$, $O_3$ |
| **Gela - Enimed** | it0815a | Italy | 37.06222 | 14.28422 | 13 | suburban background | 8052 | $NO_2$, $SO_2$, benzene |
| **Aprilia** | it0865a | Italy | 41.59528 | 12.65361 | 83 | urban background | 8169 | $NO_2$ |



| Leonessa | it0989a | Italy | 42.5725 | 12.96194 | 948 | urban background | 8207 | $NO_2$, $O_3$ |
|---|---|---|---|---|---|---|---|---|
| Gherardi | it1179a | Italy | 44.83972 | 11.96111 | -2 | rural background | 8269 | $NO_x$, $NO_2$, $O_3$ |
| Adria | it1213a | Italy | 45.04667 | 12.06194 | 4 | urban background | 8306 | $NO_2$, $NO_x$, $O_3$ |
| Cennm1 | it1375a | Italy | 39.44361 | 9.015278 | 124 | rural background | 7595 | $NO_2$, $SO_2$ |
| Teatro d'Annunzio | it1423a | Italy | 42.45639 | 14.23472 | 4 | urban background | 8135 | $NO_2$, $O_3$, $PM_{10}$, $PM_{2.5}$, $SO_2$, benzene, CO |
| Cenps7 | it1576a | Italy | 39.20333 | 8.386111 | 25 | suburban background | 7968 | CO, $NO_2$, $SO_2$ |
| Taranto San Vito | it1610a | Italy | 40.42333 | 17.22528 | 10 | urban background | 7871 | $NO_2$ |
| Lecce - S.M. Cerrate | it1665a | Italy | 40.45889 | 18.11611 | 10 | rural background | 7290 | $NO_2$, $O_3$ |
| Brindisi Via Magellano | it1702a | Italy | 40.65083 | 17.94361 | 10 | suburban background | 7904 | $NO_2$, $PM_{10}$ |
| Genga - Parco Gola della Rossa | it1773a | Italy | 43.46806 | 12.95222 | 550 | rural background | 5310 | $NO_2$, $O_3$, $PM_{10}$, $PM_{2.5}$, $SO_2$, benzene, CO |
| Civitanova Ippodromo S. Marone | it1796a | Italy | 43.33556 | 13.67472 | 110 | rural background | 6699 | $NO_2$, $NO_x$, $O_3$, $PM_{10}$, $PM_{2.5}$, benzene |
| Guardiaregia | it1806a | Italy | 41.41889 | 14.52556 | 884 | rural background | 7892 | $NO_2$, $NO_x$, $O_3$, $SO_2$ |
| Ancona Cittadella | it1827a | Italy | 43.61167 | 13.50861 | 100 | urban background | 5985 | $NO_2$, $O_3$, $PM_{10}$, $PM_{2.5}$, benzene, CO, $SO_2$ |
| Schivenoglia | it1865a | Italy | 44.99694 | 11.07083 | 16 | rural background | 8325 | $NO_2$, $NO_x$, $O_3$, $SO_2$, benzene |
| Trapani | it1898a | Italy | 38.01237 | 12.54689 | 40 | urban background | 7396 | $NO_2$, $O_3$, benzene, CO |



| | | | | | | | | |
|---|---|---|---|---|---|---|---|---|
| **San Rocco** | it1914a | Italy | 44.87306 | 10.66389 | 22 | rural background | 8398 | $NO_2$, $NO_x$, $O_3$ |
| **Locri** | it1940a | Italy | 38.22976 | 16.25518 | 11 | urban background | 8509 | $NO_2$, $O_3$, $SO_2$, benzene, CO |
| **GR - Maremma** | it1942a | Italy | 42.67056 | 11.09417 | 40 | rural background | 7784 | $NO_2$, $O_3$ |
| **Censa3** | it1947a | Italy | 39.06667 | 9.008889 | 56 | urban background | 8169 | $NO_2$, $SO_2$, benzene |
| **Milazzo - Termica** | it1997a | Italy | 38.19061 | 15.24911 | 28 | suburban background | 8329 | $NO_2$, $O_3$, CO, benzene |
| **Stadio Casardi** | it2003a | Italy | 41.31667 | 16.28611 | 15 | urban background | 8391 | $NO_2$, $O_3$, benzene |
| **Cenqu1** | it2040a | Italy | 39.23278 | 9.188056 | 8 | urban background | 8181 | $NO_2$, $O_3$, $SO_2$, benzene |
| **Carbonara** | it2051a | Italy | 41.07694 | 16.86583 | 130 | suburban background | 7505 | $NO_2$, $PM_{10}$ |
| **Cremona Gerre Borghi** | it2095a | Italy | 45.10954 | 10.06924 | 36 | rural background | 5828 | $NO_2$, $O_3$ |
| **Ceglie Messapica** | it2148a | Italy | 40.64917 | 17.5125 | 100 | suburban background | 8393 | $NO_2$, $PM_{10}$, $PM_{2.5}$, $SO_2$, CO, benzene |
| **LI - Piombino-Parco-VIII-Marzo** | it2154a | Italy | 42.93194 | 10.52417 | 40 | urban background | 8228 | $NO_2$, benzene |
| **Gela - Biviere** | it2206a | Italy | 37.02249 | 14.34497 | 0 | rural background | 8277 | $NO_2$, $O_3$, $SO_2$ |
| **Bar2** | me0008a | Montenegro | 42.10035 | 19.10348 | 12 | urban background | 7721 | CO, NO, $NO_2$, $NO_x$, $O_3$, $SO_2$ |
| **Niskic2** | me0009a | Montenegro | 42.78121 | 18.94291 | 629 | urban background | 7693 | CO, NO, $NO_2$, $NO_x$, $O_3$, $SO_2$ |
| **Celje** | si0001a | Slovenia | 46.23448 | 15.26244 | 240 | urban background | 7371 | $NO_2$, $NO_x$, $O_3$, $SO_2$ |
| **Nova Gorica** | si0034a | Slovenia | 45.95551 | 13.6524 | 113 | urban background | 8273 | $NO_2$, $NO_x$, $O_3$ |




| Koper | si0038a | Slovenia | 45.54297 | 13.71354 | 56 | urban background | 8198 | NO$_2$, NO$_x$, O$_3$ |
| Balikesir-Bandirma | tr100241 | Turkey | 40.34795 | 27.97496 | 38 | urban background | 8509 | NO$_2$ |
| Canakkale-Lapseki | tr170313 | Turkey | 40.40307 | 26.77063 | 12 | rural background | 8170 | NO$_2$, NO$_x$, O$_3$, PM$_{2.5}$, SO$_2$ |
| Istanbul-Esenyurt | tr340241 | Turkey | 41.02028 | 28.66955 | 36 | urban background | 7915 | NO$_2$, NO$_x$, SO$_2$ |
| Istanbul-Sultangazi | tr340841 | Turkey | 41.10197 | 28.87202 | 128 | urban background | 8304 | NO$_2$, NO$_x$, SO$_2$ |
| Kirkareli-Luleburgaz- | tr390441 | Turkey | 41.39841 | 27.34588 | 56 | rural background | 8393 | NO$_2$, SO$_2$ |






**Appendix C: Diurnal cycles of O₃**



**Figure C1: Diurnal cycle of O₃ in grid cells over land: (a) = Location 1, (b) = Location 2, (c) = Location 3, (d) = Location 4. Diurnal cycle of O₃ in grid cells over water: (e) = Location 5, (f) = Location 6, (g) = Location 7, (h) = Location 8. The map displays the location of the respective chosen grid cell.**



## Appendix D: NOₓ Spatial Distribution

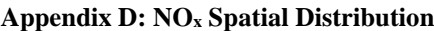

**Figure C2: Annual total dry deposition of NOₓ. (a) = CAMx, (b) = CHIMERE, (c) = CMAQ, (d) = LOTOS-EUROS. Below maps the respective frequency distribution is displayed for the annual mean NOₓ dry deposition, referred to the whole model domain.**





**Figure C3: Annual mean relative ship contribution of NO$_x$. (a) = CAMx, (b) = CHIMERE, (c) = CMAQ, (d) = LOTOS-EUROS. Below maps the respective frequency distribution is displayed for the annual mean relative ship contribution of NO$_x$, referred to the whole model domain.**




## Appendix E: annual total dry deposition of N

**Figure C4: Annual total dry deposition of N. (a) = CAMx, (b) = CHIMERE, (c) = CMAQ, (d) = LOTOS-EUROS.**