# Peer review of "Potential impact of shipping to air pollution in the Mediterranean region – a multimodel evaluation: Comparison of photooxidants $NO_2$ and $O_3$"

_EGUsphere, 2022_

## Author Comment (AC1)

**General comment**

The paper reports a modelling study of the contribution of shipping to NO2 and O3 concentrations in the Mediterranean area. In addition, deposition is included in the analysis. Results of different models are compared among themselves and with measurements in some specific stations. The topic is interesting and the paper generally well written. It has elements of novelty and I believe that it could be published after a revision step necessary to clarify some aspects and put results in a better perspective, see my specific comments.

**Specific comments**

Lines 31-32. It is not clear this sentence. The other two models use different meteorological input?

➔ These two models used the exact same input since they were run by the same institution. We added a sentence saying that the other models use different meteorological input. (*p2l33*)

Lines 41-43. I suggest to mention the recent work of Contini et al (Atmosphere 2021, 12, 92.) that gives a global overview of the effects of shipping on air quality and health.

➔ Thanks for this, we included it (*p2l46*)

Line 140. Please remove etc. if authors want to add something it is better to do it explicitly.

➔ Removed

Line 243. Actually, looking at the map in Fig. 1 it seems that it is included also the major part of eastern Mediterranean.

➔ The description is adjusted (*p10l275*)

Section 2.5. It should be mentioned how these stations have been chosen and if a certain threshold of distance from the coast or from the main routes of ships. This because it is known that the impact of the emissions from ships to air quality is strongly depending from the distance from the harbours/routes and I see some stations that are quite inland, especially in Northern Italy. A discussion on this should be provided even because I believe that the impact of shipping on such stations would be really small.

➔ No detailed distance, but we included stations with a distance of < 30 km from the coast (*p14l347*)
➔ Yes, the impact is small at inland stations, but nevertheless it makes sense to also include some stations inland to check the model performance (*p14l348*)

Another aspect that should be clarified and it is partially correlated to the previous point is if the emission dataset used include emissions of ships at berth. Several studies indicated that in EU harbours the emission at berth lead to the majority of the impact on local air quality in port cities, see for example Merico et al. (Atmospheric Environment 139 (2016) 1e10). Considering the use of low sulphur fuels at berth, this phase is particularly relevant for nitrogen oxides and could also lead to local exceedances of air quality standards. If neglected it could be present an underestimation of the impacts.

- ➔ The STEAM dataset also includes the ships at berth
- ➔ But there was no separate evaluation of harbor cities in the present study, it will be nevertheless be a part of the project to investigate harbors in city scale models

Line 336. A correlation with R=0.06 is not weak, rather it is a total absence of correlation.

- ➔ It is changed to "no to weak correlation" in the description
- ➔ It is a slightly higher value now due to new calculations which were done based on adjusted domains.
  (*p15l379*)

Lines 410-415. I believe that the results here are also comparable with those obtained with CAMx in the central/eastern part of Mediterranean area reported in Merico et al (Transportation Research Part D 50 (2017) 431–445).

- ➔ Thanks for this hint, this study is good for comparing and we now make reference of it
  (*p20l472*)

Lines 420-421. I would add or near the harbours.

- ➔ This additional information is added to the sentence
  (*p20l481*)

Figure 6. Please use the apex for m3 as in the other figures.

- ➔ The figures are adjusted
  (*Figure 3; Figure 7*)

Line 446. The absence of negative values in the tagging method is a consequence of how the method is formulated rather than a relevant result. Could this lead to problems in evaluation titration of O3?

- ➔ Tagging is no longer considered in the present study: we decided to leave this part out due to the length of the manuscript plus because it is a separate method. Furthermore, it was not goal of this study to compare the different methods.

Section 3.1.5. Regarding O3. There are several experimental evidences, some of them also in the papers that I already mentioned in my previous points, that emission of NO from ships could lead to a local reduction of O3 concentrations, especially in the spring/summer period. At larger distances instead there could be an increase. There are also some hypothesis that models could catch this behaviour more or less efficiently according to the spatial resolution of simulations. Could this be an issue in your results considering that outcomes ranging from negative to positive impacts for O3 were observed? Also Figure 16 shows that relevant differences are observed especially in the negative part.

- ➔ Yes, that explains the differences. I added these information and explained in more detail the local reduction of $O_3$
  (*p28l589*)

Line 656. I would not say air pollution considering that only NO2 and O3 are considered in this work.

- ➔ Right, this can be misleading. It is changed to "to air pollution by $NO_2$ and $O_3$"
  (*p42l741*)

Lines 714-719. This part is a little vague. It could be useful to understand if there is any possibility to understand if one of the model performs better than the other. In addition, it should be mentioned how to use results from the different models with different resolution, averaging the results?

➔ Goal of the present study was not to show if some model would perform better than another. We tried to show 1) that it is more reliable to include several model in data evaluation and 2) the range of potential impacts of ships

➔ Yes, we averaged the results in form of the boxplots (*Figure 3; Figure 7*). The resolution of 12 km and 0.1° is very similar.

---

## Author Comment (AC2)

The manuscript of Fink et al., 2022, investigates the effect of shipping emissions on NO2 and O3 in the Mediterranean Sea. Results of five different models are compared which used the zero-out method. In addition results of the tagging method in LOTOS-EUROS for NO2 are presented.

The topic of the manuscript is of high relevance as shipping emissions are an important source of pollution in the Mediterranean Sea. In the current state, however, I can't recommend a publication in ACP. To my opinion the following major points need to be adapted:

1) The biggest issue of the manuscript is the used terminology. By definition, contributions can not be calculated by the zero-out method (at least not for non-linear species), but by the LOTOS-EUROS Tagging method. I recommend to use the terminology (e.g. potential impacts and contributions) of the „Source apportionment to support air quality management practices" from FAIRMODE (https://fairmode.jrc.ec.europa.eu/document/fairmode/WG3/European%20guide%20SA_3.1_online.pdf). The clear terminology is important as the different methods (tagging, zero-out) focus on two different scientific questions. Zero-Out shows the change of e.g. ozone in case of an emission reduction. Tagging gives the contribution to ozone for the 'reference state'. Due to the different aspects of the two methods it is important to have a clear terminology to avoid any misunderstandings. A lot of literature exists on this topic for further reading, e.g.:

Wang et al., 2009, Grewe et al., 2010, Emmons et al., 2012, Butler et al., 2017, Clappier et al., 2017, Mertens et al., 2020, Belis et al., 2021, Thürkow et al, 2021, Rieger & Grewe, 2022

➔ The manuscript has been adjusted in terms of terminology, we now use "potential impact" instead of "contribution". We also removed the part of the manuscript where results achieved with the tagging method were described.

➔ We decided to not include the tagging method in the paper. The reasons are that the paper already is very long and that comparing zero-out and tagging was not main focus of this study.

➔ The zero-out method is described in more detail on *p5l144*.

2) The paper is very long. The authors tried to explain some of the (large) differences of the model results, but many differences remain unclear.

As an example, the model results for Ox look very different (e.g. compare EMEP with e.g. CAMx). Also the results for deposition differ largely (as noted by the authors), but there are no further analyses.

On p39l606 the authors note that this could be due to different dry deposition velocities, but the velocities itself are not analyzed. I understand that due to the multitude of effects and the differences between the models itself it is almost impossible to find a reason for the large spread between the models. However, in this case I suggest to reduce the amount of information (e.g. also the length) of the paper by presenting the most important findings only. This could be for example the impacts of the shipping on NO2 and O3 as simulated by the different models (and a short chapter to deposition). Also, for example, the time-series of the different models at the different stations (e.g. Figs 2 – 4) are interesting, but also very lengthy. The figures and their discussion could be moved to the supplement and 'only' the summarizing evaluation could be presented in detail.

➔ We explain the differences of $O_x$ by the different boundary conditions for $O_3$. $O_x$ pattern is very similar to $O_3$ pattern, thus also the differences from the boundary is reflected. EMEP (and CMAQ) have a lower concentration of $O_3$ coming from the eastern and southern part compared to CAMx. Further explanation was added to the manuscript and can be found: *p32l651*

➔ Information on deposition velocities are included and figures of deposition velocities are in the Supplements
(Explanations: *p36l689*; *p39l725*. Figures in Supplements: *S18, S19*)
➔ Timeseries are now moved to the appendix (*Appendix C; Appendix D*)

In addition, I suggest to better highlight/focus on what we can learn from a policy point from this study? Where are open questions? What did you learn from the multi-model study which should be considered in follow up studies? Should more things be harmonized? Where do models need to be improved?

➔ These questions are important to answer in the conclusion, you can find the answers in the last part of "6 Summary & Conclusions" (*p43/44*)
  o As policy point of the study we would see that the size of the uncertainty of single models should be considered and a model ensemble (usage of several models) could help.
  o Open questions are the potential impacts of shipping to aerosol particles and wet deposition (this is the next step of the current intercomparison study), but also detailed comparison of vertical structures and mixing of pollutants in higher layers
  o The present study has shown us possible limitations, over- and underestimations of model simulations. Limitations were traced back to the large grid sizes and the treatment of dilution of the emitted air pollutants. In addition, model-specific mechanisms lead to differences in simulation outputs.
  o Harmonization of all data was not the goal of the present study. Nevertheless, emissions were harmonized to exclude the source of uncertainty coming from emission input dataset. This was done to shed light on what other factors lead to the differences.

3) I like the idea that the different models were applied in their 'default configuration' and only resolution and anthropogenic emissions are prescribed. However, the description of the models differ strongly in their level of detail. Some examples:

- For CAMx no information about the biogenic emissions are given;

- Information about sea salt emissions are only given for CMAQ, EMEP and CAMx;

- Information about dust emissions are only given for CMAQ and EMEP.

Similarly, the description of dry- and wet-deposition differs (for example EMEP in Sect 2.1, for all other in Sect 2.4). Information about lightning NOx are missing completely.

➔ The information are adjusted and missing information was added
(see *Section 2.1*)
➔ Regarding the deposition mechanism it is all described in *Section 2.4* (*p13*)

I suggest to give the same amount of information for all models in the same level of detail. I would also suggest to expand Table 1 with details about the chemical mechanism, the used dry deposition scheme, biogenic emissions etc.

➔ Table 1 is expanded, this indeed helps to get a better overview
(Table 1 on *p5/p6*)

Finally, I further suggest to add tables with total emissions (especially for biogenic sources) for each model to the supplement. It would also be nice to see all computational domains in the supplement.

➔ There are no tables included for the emissions, since they are partly calculated on-line (i.e. biogenic emissions in CHIMERE or LOTOS-EUROS). Furthermore, this would extend the manuscript even more.

➔ Computational domains are added to the Supplements (*Supplements S1*)

In addition, I noticed that for all models the figures for NO2, O3 etc. (e.g. Fig. 7) show slightly different geographical regions. This contradicts with Fig. 1 and the information about a common domain. As example, Fig 7 (e) does only partly show the Po Valley while Fig. 7 (b), (c) and (d) show the Po Valley completely. For better comparability the same geographical region should be displayed for all models (and of course should be used for calculating mean values, frequency distribution etc.)

➔ The geographical region was adjusted and now displays the same region for all models. By correcting the region, the following things changed:
  ○ Number of measurement stations and therefore all the calculations (slightly) changed, which are connected to the stations (R, NMB etc)
  ○ Calculations between models were repeated due to the adjusted domain. Results of model intercomparison changed.

4) I suggest to replace the color scales. The rainbow color scale can be misleading. In addition it is problematic for people who are colorblind. You can check your plots for example with a 'CV Simulator' on you phone. Also some of the labels at the figures are very small. I suggest to use at leas the same font size as in the figure caption.

➔ the color scale is changed in all maps, also the font size is larger now.

Minor comments:

p6l155: Is there something missing in this sentence (boundary conditions from Mozart44 output were activated?). But more importantly, if CAMx OSAT output is available why not discuss this in the manuscript? To my opinion the paper would benefit from including OSAT results.

➔ We left out the part of the sentence where PSAT and OSAT are described because we did not consider the PSAT or OSAT results in the present study (*p7l163*)
➔ Unfortunately, the OSAT output is not available

P8l233: I am not familiar with LOTOS-EUROS, but does this mean that the model time step is 1 hour or should it read 'hourly model output' ?

➔ Maybe "hourly model outputs" fits better here for understanding (*p10l259*)

p10l257: This does mean that the NMVOC split was not adjusted to the chemical mechanisms of the individual models, right? No lumping of species were performed?

➔ We changed the description to clarify the explanation on *p12l290*

P10l264f: The part about the VOC emissions is unclear to me. Please rephrase. Thanks!

➔ We modified the explanation on *p12l299*
   o The four groups are based on how different VOC emission factors change as a function of the engine load. For example: Emission factors for VOCs in group B decrease as the engine load increases, but for VOCs in group D the emission factors will increase as the engine load increases. By dividing VOCs into these four groups, we don't have to model all species separately, but can only create emission maps for the four groups. Emissions of individual VOC species can be calculated from these emission maps. This saves computational resources needed for the modelling exercise.

P20l402: You mention the longer lifetime of NO2 for CAMx and CHIMERE. I wondered if HNO3 mixing ratios of the models differ. Please add figures in the supplement and discuss them shortly.

➔ We added the $HNO_3$ concentration and the $HNO_3$:$NO_2$ mixing ratio figures in *Supplements S11* and *S12*)
➔ A short discussion concerning $HNO_3$ can be found on *p19l448*
   o We would expect a lower $HNO_3$ concentration for models with longer lifetime of atmospheric $NO_2$. We also decided to normalized the data by using the $HNO_3$:$NO_2$ ratio. Especially along the main shipping routes differences are displayed. There, values are lower in CAMx and EMEP compared to the other models. We explained it by the lower $HNO_3$ formation by these models along the shipping routes.

Fig 6: Please don't use the tagging results for calculating mean impacts. Tagging and zero out give something different (see main point 1).

➔ We decided to not include the tagging method in the paper. The reasons are that the paper already is very long and that comparing zero-out and tagging was not main focus of this study.

p24l448: See also (1) – To my opinion the main reason zero-out gives different results (and results from different sensitivity simulations do not add up) is the non-linearity of the chemistry. Of course other factors also lead to differences.

➔ Yes, the non-linearity is one major point. Nevertheless, as we decided to not include tagging any more in this study, this point will be left out.

P26l495: Please provide figures of the different boundary conditions in the Supplement.

➔ We included information about the boundary values in the lowest model layer in the *Supplements S13* to *S16*.

P32l516ff: I don't understand this sentence. How should a split of the emissions lead to high concentrations over sea and low concentrations over land? I guess the main reason is the low dry deposition over sea, right? (as well as the overall higher land emissions).

➔ We changed this sentence to make it more clear on *p28l577*

Figure 18: The label for the subplot should be contribution frequency distribution? Please check also for all other figures.

➔ We decided to not include the labels for frequency subplots to not overload the plots

P46l679ff: Please see main point (1) above. There is also a lot of literature discussing zero out vs. tagging which could be cited here.

➔ No longer needed since tagging is no longer included (see above)

P47l691: Is there any answer on the question of and how the different dry deposition can explain the model differences?

➔ A connection can be seen between a high concentration and low deposition when the deposition velocity is low. This indicates that the substance stays longer in the atmosphere (i.e. CHIMERE). On the other hand, if the deposition rate and deposition are high, the concentration is lower (i.e. LOTOS-EUROS). (*p43l783*)

Technical comments:

I found some typos and missing spaces etc. Please double check the manuscript. Some examples:

p3l88 differences, p4l103 % by

p13l334 – Should be Table 3?

p36l580ff (and throughout the whole manuscript): I suggest to replace 'output' with model results or similar

➔ It is replaced by different terms

p46l661: The output was quantified? I guess it should read the differences of the model results was quantified or the impact of shipping simulated by the different models was quantified.

➔ Now the sentence says: "The model simulations were evaluated by comparing the simulated data against the measurements […]" (*p42l746*)

P46l673: the maps display – In my opinion the model results display (please check also the manuscript for similar wording as the term 'maps' have been used quite often)

➔ It is adjusted and other terms are being used

---

## Author Response (AR2)

1) Table 1 - Please check dust scheme for CAMx. PSAT and OSAT are the apportionment methods and not the scheme, right?

   ➔ Yes, this was a mistake, we changed it to "Calculation based on Ovadnevaite et al. (2014)" (p5)

2) I do not understand why totals of biogenic emissions can not be given. Even int he case they are calculated online information about the calculated emissions should be available from the model. Therefore, I suggest to put tables of the emission totals into the supplement.

   ➔ We have included a table of the emission totals in the supplement Table S20